

# Instantaneous aerosol and surface retrieval using satellites in geostationary orbit (iAERUS-GEO) – Estimation of 15-min AOD from MSG/SEVIRI and evaluation with reference data

Xavier Ceamanos[1], Bruno Six[2], Suman Moparthy[1*], Dominique Carrer[1], Adèle Georgeot[1], Josef
Gasteiger[3†], Jérôme Riedi[2,4], Jean-Luc Attié[5], Alexei Lyapustin[6], Iosif Katsev[7]

[1]CNRM, Météo-France, CNRS, Université de Toulouse, Toulouse, France
[2]University of Lille, CNRS, CNES, UMS 2877 – ICARE Data and Services Center, F-59000 Lille, France
[3]Faculty of Physics, University of Vienna, Vienna, Austria
[4]University of Lille, CNRS, UMR 8518 – LOA – Laboratoire d'Optique Atmosphérique, F-59000 Lille, France
[5]LAERO-Laboratoire d'Aérologie, Université de Toulouse, UPS, CNRS, 14 Avenue Edouard Belin, 31400, Toulouse, France
[6]NASA Goddard Space Flight Center, Greenbelt, Maryland, USA
[7]B.I. Stepanov Institute of Physics, National Academy of Sciences of Belarus, Pr. Nezavisimosti 68, 220072, Minsk, Belarus
[*]Now at ACRI-ST, Toulouse, France
[†]Now at Hamtec Consulting GmbH at EUMETSAT, Darmstadt, Germany

*Correspondence to*: Xavier Ceamanos (xavier.ceamanos@meteo.fr)

**Abstract.** Geostationary meteorological satellites are unique tools to monitor atmospheric aerosols from space. The observation of the Earth several times per hour allows this type of imaging systems to provide high temporal resolution observations of these suspended particles which are of interest for research topics including air quality, numerical weather prediction, and volcanic risk management. However, some challenges need to be addressed to achieve the sub-daily retrieval of aerosol properties mainly due to the varying sensitivity of geostationary imagers to aerosols during the day. In this article we propose a new algorithm named iAERUS-GEO (instantaneous Aerosol and surfacE Retrieval Using Satellites in GEOstationary orbit) that estimates the diurnal evolution of aerosol optical depth (AOD) from the Meteosat Second Generation (MSG) satellite. This is achieved by the use of an optimal estimation method combined with several aerosol models and other features including the daily retrieval of the surface reflectance directionality using Kalman filtering. AOD estimates provided by iAERUS-GEO every 15 minutes —the acquisition frequency of the Spinning Enhanced Visible Infra-Red Imager (SEVIRI) on MSG— are assessed with collocated reference aerosol observations. First, comparison to AERONET ground-based data proves the overall satisfactory accuracy of iAERUS-GEO with the exception of some higher biases found over bright surfaces and for high scattering angles. The confidence measure provided by iAERUS-GEO is proved useful to filter these less satisfactory retrievals that generally arise due to a low information content on aerosols provided by SEVIRI. Second, comparison to the GRASP/POLDER satellite product shows similar scores for the two aerosol data sets, with a significantly larger number of retrievals for iAERUS-GEO. This added value —which we illustrate here by inspecting the sub-daily variation of AOD over selected regions— allows geostationary satellites to break the temporal barrier set by traditional aerosol



remote sensing from the low Earth orbit. Furthermore, the aerosol retrievals presented in this work are expected to be improved in the near future thanks to the enhanced sensing capabilities of the upcoming Meteosat Third Generation-Imager mission.

## 1 Introduction

Aerosols consist of solid or aqueous particles of diameters in the range 0.001-10 $\mu$m that are suspended in the atmosphere and originate from a broad range of anthropogenic and natural sources. Aerosols are of the utmost importance due to their impacts on climate, weather prediction, and air quality among other key topics (Boucher, 2015). Large scale observations of aerosols —aerosol optical depth (AOD) being the most commonly retrieved variable— have been made available in the past years

thanks to remotely sensed measurements from space (Wei et al., 2020). This has been predominantly achieved with low Earth orbit (LEO) satellites located at a few hundred kilometers above the Earth's surface. This is the case of the Terra and Aqua satellites carrying the MODerate resolution Imaging Spectroradiometer (MODIS) from which a suite of well-known aerosol products is available since some years now (Hsu et al., 2013; Levy et al., 2013; Lyapustin et al., 2018).

The limited swath of sensors on LEO satellites requires a high number of orbits to reach global coverage. As a consequence,

equatorial and mid-latitude regions are generally observed one or two times per day at most. This low frequency is unfortunately not compatible with the often rapid temporal evolution of aerosols, which can travel thousands of kilometers in a few days in the instance of smoke emitted by wildfires, mineral particles in dust storms, or ashes released from volcanic eruptions. For example, Plu et al. (2021) suggested that the poor revisit time of Terra and Aqua was the reason behind the low added value of assimilating MODIS-derived AOD into the chemical transport model MOCAGE (Modélisation de la Chimie

Atmosphérique Grande Echelle) to monitor volcanic ash plumes. LEO satellites cannot capture the evolution of aerosols during the day either, which is related to a given diurnal cycle for some regions and particle types (Zhang et al., 2012). For example, Kocha et al., (2013) found that mineral dust in Northern Africa shows a decreasing or increasing diurnal cycle depending on the region and its predominant emission driver (i.e., the breakdown of the early morning low-level jet or moist convection in the afternoon, respectively). In that study, once-a-day MODIS observations could not reproduce the diurnal cycle of dust AOD

and showed mean biases with respect to model simulations ranging from -40% to +17% depending on the region and the overpass time of Terra and Aqua. Anthropogenic smoke and pollutants commonly found over urban areas also show a diurnal cycle that is driven by traffic, industrial activities, and meteorology (Backman et al., 2012). Neglect of these high frequency variations of aerosol particles can result in incorrect findings, as it was proved by Xu et al. (2016) who found an underestimation of 38.8 Wm$^{-2}$ of the daily-average direct aerosol radiative forcing over Beijing calculated using MODIS AOD

instead of sub-daily ground-based observations from the local Aerosol Robotic Network (AERONET) station.

Remote sensing of atmospheric aerosols is also accomplished from satellites in the geosynchronous equatorial orbit (GEO; also called geostationary), which are at around 36000 km of altitude and have a fixed location with respect to the Earth's surface. This type of observing systems are therefore able to acquire multiple observations of the same Earth's disk per day, approximately between two and six per hour. The potential of GEO missions to achieve high temporal resolution aerosol



monitoring was discussed in the literature. For example, Zhang et al. (2012) stated that "*the diverse patterns of aerosol daytime variation suggest that geostationary satellite measurements would be invaluable for characterizing aerosol temporal variations on regional and continental scales*" after analyzing several years of ground data at more than fifty locations across the world. Nowadays, GEO missions for Earth observation are equipped with imagers with sensing performances that are comparable to sensors on LEO satellites. This is the case of the Geostationary Operational Environmental Satellites (GOES)

from the National Oceanic and Atmospheric Administration, with GOES-16 and GOES-18 currently in operations at 75.2° W and 137.2° W, respectively (Schmit et al., 2017), the Himawari satellites from the Japan Meteorological Agency, with Himawari-9 as the present operational spacecraft at 140.7° E (Bessho et al., 2016), and the Meteosat satellites from the European Organization for the Exploitation of Meteorological Satellites (EUMETSAT), with Meteosat-11 currently being the operational satellite from the Meteosat Second Generation (MSG) program at 0° (Schmetz et al., 2002).

Retrieval algorithms have been developed in the past years to monitor aerosols from GEO satellites including MSG (Govaerts et al., 2010; Luffarelli and Govaerts, 2019; Thieuleux et al., 2005), GOES (Knapp, 2002; Kondragunta et al., 2020), Himawari (Gupta et al., 2019; Lim et al., 2018; Yoshida et al., 2018) and other missions such as the Korean Geostationary Ocean Color Imager (GOCI; Choi et al., 2016). One example is the AERUS-GEO (Aerosol and surface albEdo Retrieval Using a directional Splitting method-application to GEOstationary data) method which provides AOD at 635 nm from the Spinning Enhanced

Visible Infra-Red Imager (SEVIRI) on MSG (Carrer et al., 2010; 2014), and more recently from a constellation of GEO imagers providing quasi-global coverage (Ceamanos et al., 2021). One of the main strengths of this algorithm is the use of a Kalman filter to estimate the surface bidirectional reflectance distribution function (BRDF) —a key parameter for a successful aerosol retrieval— and to propagate it with time such that it can be used as prior information in future days. This approach, which exploits the slower evolution of surface properties with respect to aerosols', provides estimates of AOD that were proved

useful in several studies (Escribano et al., 2017; Roberts et al., 2018; Xu et al., 2014). Nonetheless, AERUS-GEO retrieves AOD at the daily frequency only, by simultaneously processing all valid satellite measurements recorded during the day, thus not exploiting the high frequency of SEVIRI with one full Earth's disk image every 15 minutes.

This work overcomes this limitation with the new algorithm named iAERUS-GEO (instantaneous Aerosol and surfacE Retrieval Using Satellites in GEOstationary orbit) that performs *instantaneous* estimation (i.e., at the imager acquisition

frequency) of aerosol load from geostationary meteorological satellites. The application of this method to SEVIRI/MSG with the purpose of providing maps of AOD at 635 nm every 15 minutes is presented here. iAERUS-GEO inherits some concepts from the original AERUS-GEO algorithm such as the daily estimation of surface BRDF with a Kalman filter. However, many significant changes were introduced to address the challenges of estimating the diurnal evolution of AOD that arise from the drastically reduced number of satellite measurements available for instantaneous inversion and the need for accurate radiative

transfer calculations across the broad range of geometries made available by GEO satellites. The latter point is crucial as the sensitivity to aerosols varies significantly during the day for this type of imaging systems (Luffarelli and Govaerts, 2019; Ceamanos et al., 2019). In this context, some approximations made in AERUS-GEO that were acceptable for daily retrieval had to be revisited or abandoned for iAERUS-GEO. These include the use of a simplified radiative transfer model (RTM)





neglecting the anisotropy of ocean reflectance and considering a sole non-absorbing aerosol model described by a single-lobed
Henyey-Greenstein phase function. All limitations were overcome in iAERUS-GEO, which performs optimal estimation of
AOD over land and ocean —considering the specific anisotropy of each surface type— based on a multi-pixel technique and
a set of aerosol models representative of the diverse atmospheric particles found around the world. In addition, an efficient
RTM is used to perform calculations with a good trade-off between precision and speed, which allows iAERUS-GEO to
process GEO data in near real time.

This article is organized as follows. The iAERUS-GEO algorithm is described in Sect. 2 and the experiments that were
conducted to assess its accuracy are detailed in Sect. 3. Results are reported in Sect. 4 and conclusions are drawn in Sect. 5.

## 2 Retrieval algorithm

### 2.1 Overview

iAERUS-GEO retrieves AOD at 635 nm from the cloud-free pixels of each SEVIRI image. This results in the estimation of a
map of AOD across the MSG Earth's disk every 15 minutes during daytime. The following data are required as input:

- Full-disk images of top-of-atmosphere (TOA) reflectance from channel VIS06 centered at 635 nm and corresponding
  to the shortest wavelength available on SEVIRI.
- Solar and view zenith and azimuth angles.
- Static files including maps of latitude and longitude, a land-water mask, a mask for coastal pixels, and a digital
elevation model.
- Binary cloud mask to limit the processing to cloud-free pixels only.
- Fields of surface pressure, total column water vapor, and total column ozone to perform molecular correction.
- Fields of surface wind speed and direction to calculate the reflectance of ocean surfaces.
- Auxiliary data on aerosols including:
- Climatological monthly AOD averages to be used as prior information.
  - Optical properties for a set of seven aerosol models.
  - Monthly maps giving the geographic distribution of aerosol models.

Figure 1 summarizes the retrieval process performed in iAERUS-GEO, which is composed of three main steps:

1. Correction for molecular effects executed for cloud-free pixels of every satellite image (Sect. 2.3).
2. Estimation of the surface BRDF to characterize the reflectance directionality, executed at the end of the day using all
   available satellite images (Sect. 2.4).
3. Estimation of instantaneous AOD, executed for every satellite image individually (Sect. 2.5).



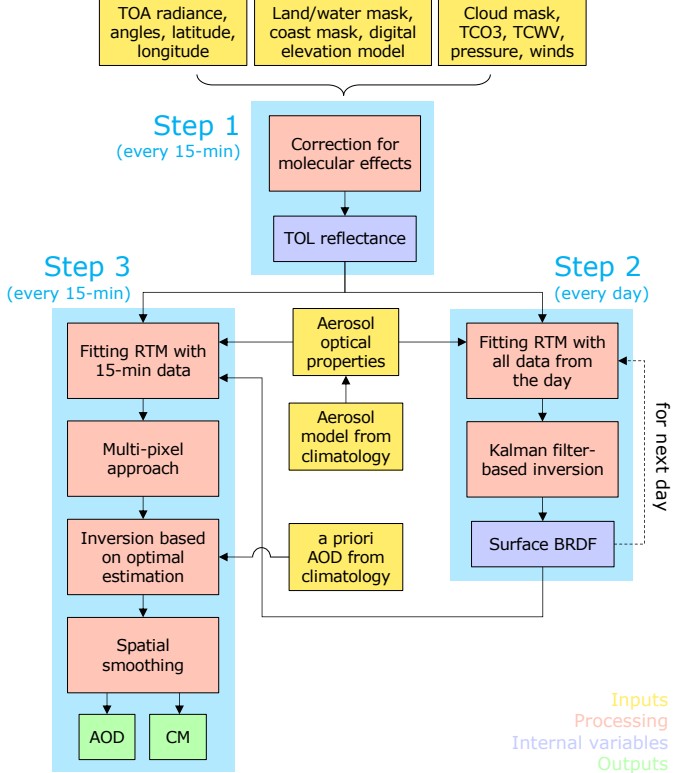

**Figure 1.** Schematic view of iAERUS-GEO. TOL reflectance is defined at the top of the aerosol layer after gas correction. CM is the confidence measure provided along with each AOD estimate. TCO3 and TCWV are the total ozone and water vapor columns, respectively.

Steps 2 and 3 are executed with the analytical RTM described in Sect. 2.2, and the auxiliary aerosol data (Sect. 2.6). These two processing steps are nested as shown in Fig. 2 to fulfill the need for surface reflectance in the estimation of AOD. Each full-disk image is processed individually (solid arrows) to retrieve instantaneous AOD (blue boxes) for each pixel using the latest available surface BRDF (green boxes). The latter parameter is updated daily for each pixel by the use of all the available diurnal measurements (dashed arrows). Surface BRDF is propagated with time to be used as prior information in the daily inversion of the next day. This strategy has two main advantages. First, it allows the consideration of the bidirectional effects of surface reflectance by estimating BRDF instead of individual reflectance values as it is done in other algorithms (e.g., Yoshida et al., 2018). Second, it allows iAERUS-GEO to satisfy the constraints of near real time processing that make impossible the use of the surface BRDF of the current day. The use of past estimates relies on the assumption of invariability of the directionality of surface reflectance for a time offset of a few days (Carrer et al., 2010; Lyapustin et al., 2018).

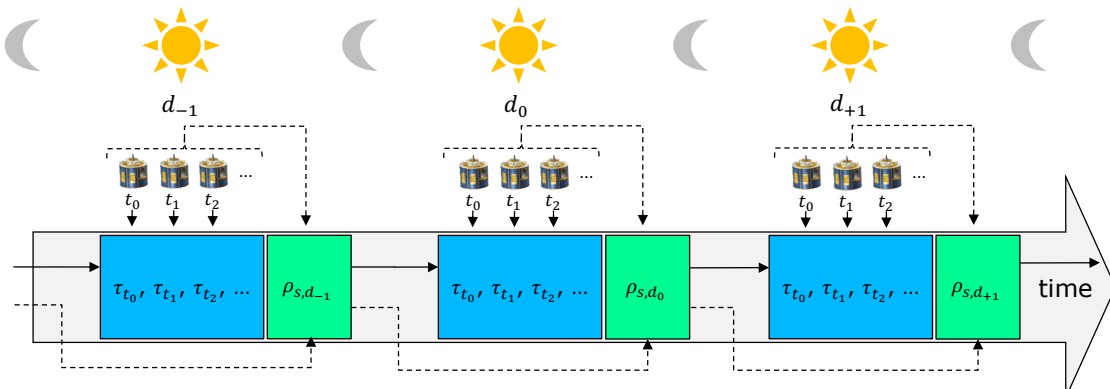

**Figure 2.** Nesting of daily and instantaneous processing during the days ($d_k$). Daily retrieval of surface BRDF ($\rho_s$) is shown with dashed lines and green color boxes. Instantaneous retrieval of AOD ($\tau$) at every time slot ($t_j$) is illustrated with solid arrows and blue color boxes.

The retrieval process in iAERUS-GEO slightly differs for land and ocean surfaces due to their distinct reflectance characteristics. The main difference is that appropriate models of BRDF are used for each type of surface (Sect. 2.2.3), which is determined for every SEVIRI pixel with the land-water mask. Other differences are explained in the following sections. Coastal pixels, usually containing both ocean and land, are not processed until the last step described in Sect. 2.5.4.

## 2.2 Radiative transfer model

### 2.2.1 Atmosphere model and expression for top-of-layer reflectance

In iAERUS-GEO, the Earth's atmosphere is assumed to consist of two layers (Fig. 3a). The upper layer is composed of gas molecules and the lower layer, right above the surface, is made of a mixture of aerosol particles and gases. This simple yet efficient model has already been used successfully in the literature (e.g., Govaerts et al., 2018; Katsev et al., 2010).

The first step of iAERUS-GEO is the compensation of TOA reflectance ($\rho_{TOA}$) satellite measurements for molecular effects including gas absorption and Rayleigh scattering (quantified by its optical depth $\tau_r$). This results in values of top-of-layer (TOL) reflectance ($\rho_{TOL}$), which is defined at the top of the aerosol layer and depends on the contributions from the surface and the aerosols only. Details on this correction and the underlying hypotheses are given in Sect. 2.3. In the absence of gases, the atmosphere can be represented with a single layer in which the extinction of solar radiation exclusively comes from aerosols (Fig. 3b). Analogously to the well-known expression from Chandrasekhar (1960), TOL reflectance can be written as

$$\rho_{TOL}(\theta_s, \theta_v, \varphi) = \rho_{aer}(\theta_s, \theta_v, \varphi) + \frac{T_{aer}^\downarrow(\theta_s) T_{aer}^\uparrow(\theta_v)}{1 - a_{aer} a_s} \rho_s(\theta_s, \theta_v, \varphi), \tag{1}$$

where $\rho_{aer}$ is the aerosol layer reflectance, $T_{aer}^\downarrow$ is the downwelling aerosol transmittance, $T_{aer}^\uparrow$ is the upwelling aerosol transmittance, $a_{aer}$ is the spherical (or bi-hemispherical) albedo of the aerosols at illumination from bottom upwards, $\rho_s$ is the bidirectional surface reflectance, and $a_s$ is the spherical albedo of the surface. All aerosol terms depend on the aerosol optical depth ($\tau_{aer}$), which will be referred to $\tau$ hereafter for the sake of simplicity. The view and solar geometry are defined by the




solar zenith angle $\theta_s$, the view zenith angle $\theta_v$, and relative azimuth angle $\varphi$ (obtained from its solar and view counterparts
making $\varphi = \varphi_s - \varphi_v$). Another important angle in aerosol remote sensing is the scattering angle $\xi$ that is calculated as

$$\xi = \pi - \arccos(\cos\theta_s \cos\theta_v + \sin\theta_s \sin\theta_v \cos\varphi), \tag{2}$$

with $\xi = 0$ corresponding to the forward direction, with the Sun in front of the sensor and when aerosol scattering is maximum,
and $\xi = \pi$ corresponding to the backward direction, with the Sun behind the sensor and when aerosol scattering is lower.

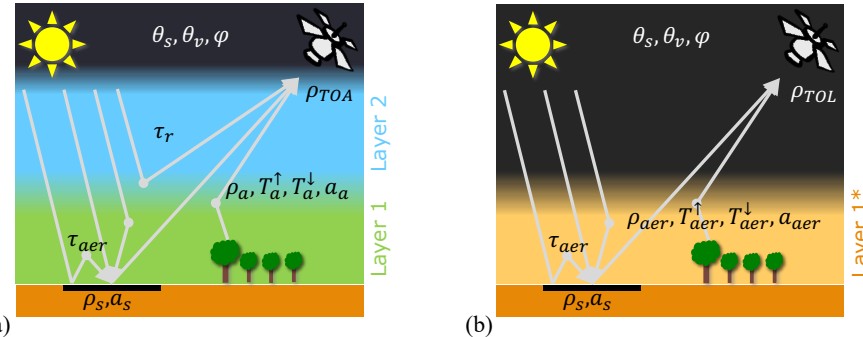

**Figure 3.** Atmosphere model and radiative parameters (a) before and (b) after correction for gas effects. Subscripts "aer" and "a" are
respectively used to distinguish between terms related to aerosols or to the whole atmosphere (aerosols plus gases).

**2.2.2 Solution for aerosol contribution using the Modified Sobolev's Approximation**

Aerosol terms in Eq. 1 are calculated with the Modified Sobolev's Approximation (MSA) proposed by Katsev et al. (2010).
This model provides approximate analytical functions by combining the Sobolev's approximation (Sobolev, 1975) with a
truncated phase function, thus allowing fast calculations of TOL reflectance. MSA describes the aerosol layer through the
optical properties $P(\xi)$, $\tau$, $\omega$, $g$ (i.e., the scattering phase function, the AOD, the single scattering albedo, and the asymmetry
parameter, respectively) and its reflectance is modeled as the sum of single scattering (SS) and multiple scattering (MS)

$$\rho_{aer} = \rho_{aer}^{SS} + \rho_{aer}^{MS}. \tag{3}$$

More details on MSA and all the equations for aerosol reflectance and transmittance terms are given in Appendix A.1.
According to Katsev et al. (2010), MSA allows the estimation of the aerosol layer reflectance with an error lower than 10%
except for high AOD and high zenith angles. These reasonable yet perfectible performances are counterbalanced by the high
speed of MSA, which is crucial for aerosol retrieval from GEO platforms in near real time, with more than 10 million pixels
to process in less than 15 minutes in the instance of SEVIRI. Katsev et al. (2010) used a truncating angle ($\xi^*$) equal to 45°,
which resulted in the discard of the data from the LEO satellite Envisat used in their study with a scattering angle lower than
that value. Here, a value of $\xi^* = 30°$ is chosen instead to account for the generally lower scattering angles reached by GEO
satellites. This truncation angle was found to provide a similar accuracy of MSA than the one reported in Katsev et al. (2010)
according to numerical calculations performed with the radiative transfer code libRadtran (Emde et al., 2016) (not shown here).



### 2.2.3 Solution for surface contribution using a kernel-based BRDF model approach

TOL reflectance depends on the bidirectional reflectance of the surface $\rho_s$ (Eq. 1), which often shows a significant diurnal
variation in GEO observations due to varying solar geometry and reflectance anisotropy. This directionality is taken into
account in iAERUS-GEO through the use of semi-empirical BRDF models based on the linear combination of $n$ functions
(i.e., kernels) corresponding to different scattering processes

$$\rho_s = \boldsymbol{kf} = \sum_{i=0}^{n-1} k_i f_i, \tag{4}$$

where coefficients $k_i$ are initially unknown, as they depend on the properties of the region observed at each pixel, and functions
$f_i$ are known, as they depend on ancillary information such as solar and view geometry.

Different models are used over land and ocean to account for the distinct anisotropy of these two types of surfaces. First, the
hotspot-corrected Ross-Li model (Maignan et al., 2004) is used to represent surface BRDF for land surfaces. The land surface
reflectance ($\rho_s^l$) is expressed as the sum of $n=3$ kernels ($f_i^l$) describing isotropic, geometric, and volumetric processes.
Expressions for $f_i^l$ depend on geometry only and are given in Appendix A.2.1. Second, the model used for water surfaces
follows the work from Koepke (1984), who defines ocean reflectance ($\rho_s^o$) as the sum of three contributions (i.e., whitecaps,
sun glint, and underlight). According to Sayer et al. (2010), among the three terms only sun glint shows a significant directional
variability. Hence, surface BRDF over ocean is expressed here as the linear combination of $n=2$ functions ($f_i^o$) describing
isotropic (from whitecaps and underlight) and anisotropic (from sun glint) contributions. Expressions for $f_i^o$ depend on
geometry and surface winds, and are given in Appendix A.2.2.

**2.3 Correction for molecular effects**

Cloud-free pixels are corrected for gas absorption and Rayleigh scattering using the Simplified Model for Atmospheric
Correction (SMAC; Rahman and Dedieu, 1994). SMAC performs a fast correction for several atmospheric gases including
$O_2$, $CO_2$, $H_2O$, and $O_3$ based on parametric equations fitted with radiative transfer simulations. Gases are set to the
concentration values defined in the U.S. Standard Atmosphere model except for ozone and water vapor, which are quantified
with model analyses used as input by SMAC. Surface pressure is also required as input to account for the variation of gas
effects with surface height, which is derived from a digital elevation model. The accuracy of SMAC is within 2-3%, if slope
effects are mild and high viewing and solar angles are avoided.

Correction for gas effects is done by ignoring the coupling between molecular and particular scattering. This hypothesis is
found to be reasonable for channel VIS06 according to Rozanov and Kokhanovsky (2005), who found this coupling to be
negligible (<1%) for wavelengths greater than 600 nm. In practice, SMAC calculates $\rho_{TOL}$ making $\tau = 0$, which allows the
subtraction of the terms related to gases from the values of TOA reflectance. This processing step is inherited from the original
AERUS-GEO algorithm but was recently updated by recalculating the fitting coefficients used in SMAC based on simulations
from the code 6SV1 (Kotchenova et al., 2006), which includes few improvements with respect to the previously used code 6S.





### 2.4 Daily retrieval of surface BRDF

#### 2.4.1 Inversion method and Kalman filtering

Surface BRDF is estimated at the end of the day for each SEVIRI pixel by the use of all the available observations of $\rho_{TOL}$. This is done following a strategy that is similar to the one used in the original AERUS-GEO algorithm (Carrer et al., 2010) to retrieve daily-average AOD ($\tau_{\text{daily}}$) and surface BRDF ($\rho_s$) simultaneously. Modifications were made in the instance of iAERUS-GEO to provide the best surface BRDF possible —contrary to AERUS-GEO which focuses on $\tau_{\text{daily}}$, being the main

output— as a reliable estimate of surface reflectance is key to achieve the instantaneous estimation of AOD during the day. The daily inversion exploits the linearity of the $\rho_{TOL}$ expression in Eq. 1, after combining it with Eq. 3, and after introducing the kernel-based expressions for the surface BRDF in Eq. 4 and an extra kernel for the aerosol single scattering reflectance

$$\rho'_{TOL} = \rho_{TOL} - \rho_{aer}^{MS} = \boldsymbol{k}\boldsymbol{f}' = \sum_{i=0}^{n} k_i f_i'. \tag{5}$$

The resulting linear system has $n+1$ kernels (i.e., 4 for land pixels and 3 for ocean pixels), with the state parameters defined

by the vector $\boldsymbol{k}$ that is equal to $\left[k_0^l, k_1^l, k_2^l, \tau_{\text{daily}}\right]$ for land and $\left[k_0^o, k_1^o, \tau_{\text{daily}}\right]$ for ocean. AOD is assumed to be constant throughout the day to reduce the aerosol parameters to one. The expressions for the vector of modified kernels $\boldsymbol{f}'$ (depending on $\boldsymbol{f}$ in the instance of surface BRDF, Eq. 4) are given in Appendix B.

Inversion is done based on the Kalman filtering theory that uses the satellite observations and the previous surface solution

$$\boldsymbol{k} = \frac{A^T b + C_{ap}^{-1} \boldsymbol{k}_{ap}}{C_k^{-1}}, \tag{6}$$

with the associated covariance matrix $\boldsymbol{C}_k$

$$\boldsymbol{C}_k = \left(\boldsymbol{A}^T \boldsymbol{A} + \boldsymbol{C}_{ap}^{-1}\right)^{-1}. \tag{7}$$

The kernel matrix $\boldsymbol{A}$ is defined with the elements $A_{ij} = f_{ij}'\vartheta_j$, where $j$ and $i$ refer to the different observations and kernels, respectively. The data vector $\boldsymbol{b}$ is composed of the scaled satellite observations with the elements $b_j = \rho'_{TOL,j}\vartheta_j$. The weighting factors $\vartheta_j$ give greater importance to certain observations according to their angular characteristics as detailed in Sect. 2.4.2.

The surface solution corresponding to the previous day ($d_{-1}$) is used as prior information in the inversion by making $\boldsymbol{k}_{ap} = \boldsymbol{k}^{d_{-1}}$. Analogously, the covariance matrix is also propagated in time by making

$$\boldsymbol{C}_{ap} = \boldsymbol{C}_k^{d-1}\boldsymbol{\delta}^{age}, \tag{8}$$

where the multiplying term is used to modulate the weight of the a priori information. First, vector $\boldsymbol{\delta} = [\delta_0, \delta_1, \dots]$ is used to impose the distinct temporal variability of each surface kernel with $\delta_i = 2^{2/t_i}$. For land, the values $t_i = [10,60,60]$ were

chosen to impose a lower variation of the directionality of the surface BRDF (i.e., $k_1^l$ and $k_2^l$) whereas the isotropic contribution (i.e., $k_0^l$) is allowed to vary faster to account for rapid variations of reflectance (e.g., due to rainfall). For ocean, the values $t_i = [60,60]$ were found to provide satisfactory results. In the two cases the greatest part of the daily variations in satellite





observations are assigned to aerosol variability by not constraining the aerosol kernel (i.e., $\tau_{\text{daily}}$ is assumed to be independent from one day to another). Second, exponent $age$ —the number of days since the a priori surface BRDF was updated— is used

to decrease the weight of "old" prior information. Note that successful daily inversions make $age = 0$ whereas unsuccessful retrievals (e.g., due to the presence of clouds) result in $age = age^{d-1} + 1$. In the latter case the previous surface solution is propagated in time making $\boldsymbol{k} = \boldsymbol{k}^{d-1}$ and $\boldsymbol{C}_k = \boldsymbol{C}_k^{d-1}$.

Additional conditions are imposed to obtain the best possible surface BRDF. First, a minimum of three hours of valid satellite observations is required to avoid poorly constrained surface solutions. Second, surface BRDF is only updated if the retrieved

$\tau_{\text{daily}}$ is lower than 1 to avoid potential spurious aerosol contamination. Third, climatologic values of AOD (Sect. 2.6.1) are used as solution for $\tau_{\text{daily}}$ to ease the estimation of surface BRDF when simultaneous aerosol-surface estimation becomes difficult (e.g., over bright surfaces). The resulting Kalman filter-based approach results in temporally smooth surface estimates after few days of processing in most of cases thanks to the continuous flow of data provided by GEO satellites.

**2.4.2 Double inversion approach**

The estimation of surface BRDF may not be straightforward due to the broad sampling in solar angles of GEO measurements. Furthermore, kernel-driven BRDF models such as Ross-Li's were reported to show limitations in representing the whole range of zenith angles (Zhang et al., 2018). These issues are circumvented in iAERUS-GEO by estimating surface BRDF twice, one for the backward hemisphere and one for the forward hemisphere. A first inversion of all valid observations is done with a set of weights ($\vartheta^1$, Sect. 2.4.1) that decrease with the measurement scattering angle (pink color line in Fig. 4). This inversion

provides the first estimate ($\rho_s^1$, through the estimation of the corresponding surface coefficients $k_i^1$) together with an estimate of $\tau_{\text{daily}}$. This is generally possible as greater weights are given to observations for which aerosol scattering is maximum. A second inversion is done with another set of weights ($\vartheta^2$) that increase with scattering angle (blue color line in Fig. 4). The value of $\tau_{\text{daily}}$ found in the first inversion is used here to provide the second estimate ($\rho_s^2$, by means of $k_i^2$).

Instantaneous retrieval of AOD uses the surface reflectance resulting from the weighted combination of the two estimates

$$\rho_{s,j} = \left( \vartheta^1(\xi_j)\rho_{s,j}^1 + \vartheta^2(\xi_j)\rho_{s,j}^2 \right) \left( \vartheta^1(\xi_j) + \vartheta^2(\xi_j) \right)^{-1}, \tag{9}$$

with $\xi_j$ corresponding to the scattering angle of the satellite observation $j$. Experiments showed the benefits of combining the two surface BRDF estimates with respect to the use of one estimate only.

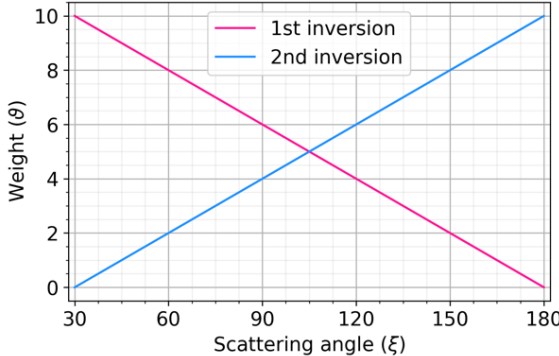

**Figure 4.** Weighting factors for the inversion of the two estimates of surface BRDF, starting at the truncating angle $\xi^* = 30°$.

**2.5 Instantaneous retrieval of AOD**

**2.5.1 Multi-pixel technique**

AOD is estimated from each available $\rho_{TOL}$ observation, at the SEVIRI frequency of 15 minutes. Retrieval is performed based on the pixel to be processed and its adjacent pixels in space and time. This is done with a multi-pixel technique inspired by earlier works that assume that AOD is constant at small spatial and temporal scales (e.g., Katsev et al., 2010; Dubovik et al.,

2011; Shi et al., 2019). In iAERUS-GEO, the same value of AOD is assumed within the spatiotemporal super-pixel defined by the 3 × 3 pixel box centered on the pixel to be processed and spanning from the acquisition time ($t_0$) to 2 hours before (Fig. 5a). Darker pixels are given a greater weight in the retrieval of AOD as it is done in the Dark Target algorithm (Levy et al., 2013). The size of super-pixels for SEVIRI (9 × 9 km over the subsatellite point and around 15 × 15 km over Western Europe) is in agreement with Anderson et al. (2003), who found that AOD does not vary significantly for time (space) offsets of 3

hours (60 km) based on ground observations. Each super-pixel with $\boldsymbol{\rho_{TOL}} = \{\rho_{TOL}^1, \rho_{TOL}^2, ...\}$ is processed as follows:

1. Ocean (land) pixels are removed when the pixel to be processed is over land (ocean).

2. Very bright pixels are discarded by filtering $\rho_{TOL}^j$ values showing a deviation greater than one sigma from the average.

3. Remaining pixels are averaged with two sets of weights making $\overline{\rho_{TOL}} = \sum_i \rho_{TOL}^i \gamma_\rho^i \gamma_t^i$. Weights $\gamma_\rho^i$ are defined according to the value of TOL reflectance following Fig. 5b, which gives greater weights to darker pixels. Weights

$\gamma_t^i$ give lower weights to past observations following the function in Fig. 5c, which was derived from the work of Anderson et al. (2003) (Fig. 6 of that study).

4. Surface reflectance ($\boldsymbol{\rho_s}$) and surface albedo ($\boldsymbol{a_s}$) values are averaged following the same approach.

The obtained mean values of $\overline{\rho_{TOL}}$, $\overline{\rho_s}$, and $\overline{a_s}$ are ascribed to the pixel to be processed for AOD retrieval.



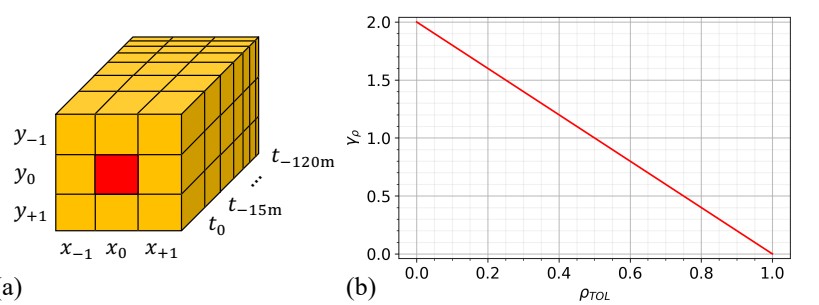

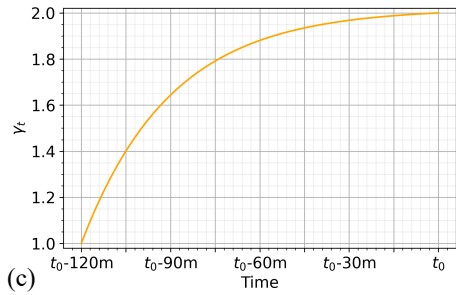

(a)  (b)  (c)

**Figure 5.** (a) Scheme of super-pixel with pixel to be processed in red color, (b) Weights depending on TOL reflectance when $\max(\boldsymbol{\rho_{TOL}}) = \mathbf{1}$ and $\min(\boldsymbol{\rho_{TOL}}) = \mathbf{0}$, and (c) weights depending on acquisition time.

### 2.5.2 Inversion method

AOD at 635 nm is estimated by the use of the optimal estimation theory (Rodgers, 2000). This approach provides a solution that optimizes the balance between the estimation that can be achieved from the satellite data and the one coming from prior information. At the $i + 1^{\text{th}}$ iteration, the previous estimate ($\tau_i$) is updated with the Levenberg-Marquardt equation as

$$\tau_{i+1} = \tau_a + \left(K_i S_y^{-1} K_i + (1+\gamma)S_a^{-1}\right)^{-1}\left(K_i S_y^{-1}\left(\rho_{TOL} - \hat{\rho}_{TOL}(\tau_i) + K_i(\tau_i - \tau_a)\right) + \gamma S_a^{-1}(\tau_i - \tau_a)\right), \tag{10}$$

where $\tau_a$ is the a priori AOD with its corresponding error variance $S_a$, $\rho_{TOL}$ is the TOL reflectance measured by the satellite at 635 nm with its corresponding error variance $S_y$, $\hat{\rho}_{TOL}(\tau_i)$ is the equivalent TOL reflectance calculated with the RTM described in Sect. 2.2, and $K_i$ is the AOD Jacobian of $\hat{\rho}_{TOL}$ at iteration $i$. In iAERUS-GEO, $\tau_a$ comes from a model-based monthly climatology (Sect. 2.6.1). After many experiments $S_a$ was set to $0.05^{(1+\rho_s)}$ —giving a greater weight to the prior information for retrievals over bright surfaces— whereas $S_y$ was set to 0.0001.

Finally, $\gamma$ is a parameter that is adjusted at each iteration to minimize the cost function defined as

$$\chi_i^2 = (\tau_i - \tau_a)^2 S_a^{-1} + \left(\rho_{TOL} - \hat{\rho}_{TOL}(\tau_i)\right)^2 S_y^{-1}. \tag{11}$$

At the first iteration, the inversion starts with $\gamma = 1$ and $\tau_0 = \tau_a$. If the $\chi_i^2$ value calculated at iteration $i$ decreases, we reduce $\gamma$ by a factor of 2 and we move to iteration $i + 1$. Conversely, if the $\chi_i^2$ value increases, we increase $\gamma$ by a factor of 2 and we repeat the iteration. In our case the calculation stops after 8 iterations, which corresponds to the maximum number of iterations needed to reach the convergence of the system.

### 2.5.3 Confidence measure

AOD estimates are provided with an indicator of their robustness by means of a confidence measure (CM). This output parameter of iAERUS-GEO is calculated based on the sensitivity of satellite measurements to AOD (i.e., their information content), which is quantified by the absolute value of the AOD Jacobian $|K|$ after the last iteration of the inversion. As seen in Fig. 6, lower values of CM are given to AOD estimates as $|K|$ gets close to 0, which corresponds to the situation when satellite





measurements are insensitive to aerosol load and retrievals are more likely to be unreliable. For the same range of $|K|$, CM is one unit lower for bright surfaces (i.e., $a_s > 0.2$) to account for the general greater uncertainty of AOD retrievals in this case.

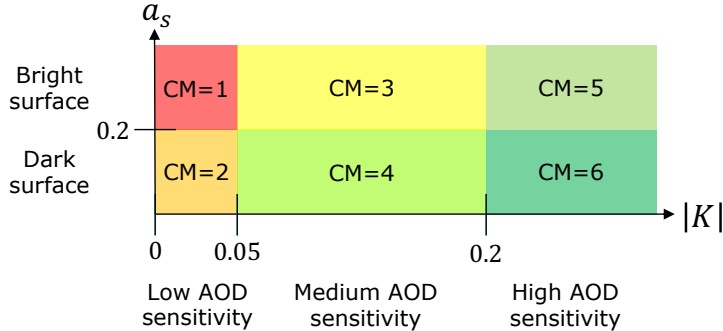

**Figure 6.** Definition of the CM provided with each AOD estimate based on the absolute AOD Jacobian $|K|$ and the surface albedo $a_s$.

### 2.5.4 Spatial smoothing

Each 15-min map of retrieved AOD is spatially improved following Lyapustin et al., (2018). Three steps are performed:

1. AOD maps are filtered with a $3 \times 3$ -pixel running window that removes excessively high AOD values with respect to adjacent pixels. In particular, the maximum AOD in the window ($\tau_{\max}$) is filtered if $\tau_{\max} > \tau_{\mathrm{avg}} + 0.15$, where $\tau_{\mathrm{avg}}$ is the average of the window computed without $\tau_{\max}$.

2. Coastal pixels are filled with the result of a $9 \times 9$ -pixel running averaging window. The larger window size is justified by the low number of AOD estimates that are available for averaging, as coast pixels are not processed until here.

3. A final $3 \times 3$ -pixel running averaging window is applied to the resulting map of AOD.

This spatial smoothing step reduces residual errors based on the assumption that aerosols are spatially homogeneous.

## 2.6 Auxiliary aerosol data

Ancillary data on aerosols are used in iAERUS-GEO for two purposes. First, the optimal estimation of instantaneous AOD is performed using a priori values of aerosol load coming from a model-based climatology (Sect. 2.6.1). Second, radiative transfer calculations are done considering a specific aerosol model, out of seven available (Sect. 2.6.2), that is determined for each SEVIRI pixel based on a set of monthly geographic distribution maps (Sect. 2.6.3).

### 2.6.1 Monthly maps of a priori AOD

Values of a priori AOD are obtained from the 11-year reanalysis of atmospheric composition produced by the Copernicus Atmosphere Monitoring Service (CAMS, Bozzo et al., 2020). This climatological data set was used to calculate monthly averages of total AOD by considering the contribution of each of the five species considered in CAMS: sea salt (SS), dust (DU), organic matter (OM), black carbon (BC), and sulfate (SU). The resulting data sets of total AOD ($AOD_{\mathrm{total}}$), and those corresponding to each species ($AOD_{SS}$, $AOD_{DU}$, $AOD_{OM}$, $AOD_{BC}$, $AOD_{SU}$), were projected onto the SEVIRI grid.



### 2.6.2 Aerosol models

Seven models are used in iAERUS-GEO to represent the large variety of atmospheric particles on Earth. This includes six types of aerosols originated over land and one having its origin over ocean. Land aerosol models are borrowed from the Multi-
Angle Implementation of Atmospheric Correction (MAIAC) C6 algorithm applied to MODIS (Lyapustin et al., 2018) including two continental types —Model 1 representative of Eastern USA with high summertime humidity and Model 4 representative of Europe with higher absorption—, one arid climate type —Model 2 representative of Western USA with larger coarse fraction due to dust particles—, one polluted type —Model 8 representing industrial India with high absorption due to agricultural biomass burning and transportation—, a desert dust type —Model 6 made of non-spherical mineral particles—,
and a biomass-burning type —Model 7 representing subequatorial Africa—. The maritime aerosol model (Model 0) represents sea salt particles found over the Atlantic Ocean.

Models are representative of the aerosol climatology of the corresponding regions and were adjusted with observations of selected AERONET sites. Over land, for example, Models 6 and 7 were tuned by Lyapustin et al. (2018) based on the Solar Village and Mongu sites, respectively. Over ocean, Model 0 was built by averaging the microphysical properties reported by
Sayer et al. (2012) for the sites in Ascension Island, Graciosa, and Bermuda. Models can be either static (Models 0 and 2) with fixed parameters or dynamic (Models 1, 4, 6, 7, and 8) with parameters depending on AOD to represent variations in particle sizes and in the ratio of fine to coarse modes (Remer and Kaufman, 1998). Parameters are given in Table 1 for all models.

**Table 1.** Microphysical properties of aerosol models considered in iAERUS-GEO: radius and standard deviation of fine and coarse fractions of bi-lognormal volume size distribution; ratio of volume concentrations (coarse to fine) as functions of AOD; real and imaginary refractive
index at 635 nm ($n = m - ik$). For Model 0, refractive index is different for fine and coarse particles, with the values in parentheses corresponding to the latter particles. Last column shows the fraction of spherical particles with respect to spheroids.

| Model | Type | $R_v^F$ | $\sigma_v^F$ | $R_v^C$ | $\sigma_v^C$ | $C_v^C/C_v^F$ | $m$ | $k_{0.635}$ | Mie fraction |
|---|---|---|---|---|---|---|---|---|---|
| 0 | Maritime | 0.1647 | 0.557 | 2.433 | 0.74 | 4.37 | 1.415 (1.363) | 0.002 (0.000) | 1 |
| 1 | Continental USA | $0.12 + 0.05\tau$ $\leq 0.2$ | $0.35 + 0.05\tau$ $\leq 0.45$ | $2.8 + 0.2\tau$ $\leq 3.2$ | $0.6 + 0.1\tau$ $\leq 0.8$ | 0.6 | 1.42 | 0.0045 | 1 |
| 2 | Arid climate | 0.16 | 0.4 | 2.4 | 0.6 | 0.5 | 1.48 | 0.0035 | 0.8 |
| 4 | Continental Europe | $0.12 + 0.05\tau$ $\leq 0.2$ | $0.35 + 0.05\tau$ $\leq 0.45$ | $2.8 + 0.2\tau$ $\leq 3.2$ | $0.6 + 0.1\tau$ $\leq 0.8$ | 0.6 | 1.42 | 0.0065 | 1 |
| 6 | Desert dust | 0.12 | 0.5 | 1.9 | 0.6 | $\dfrac{0.9\tau}{0.02(1+\tau)}$ | 1.56 | 0.0011 | 0 |
| 7 | Biomass burning | $0.12 + 0.025\tau$ $\leq 0.2$ | 0.4 | $3.2 + 0.2\tau$ $\leq 3.8$ | 0.7 | 0.7 | 1.51 | 0.009 | 1 |
| 8 | Polluted India | $0.15 + 0.05\tau$ $\leq 0.2$ | $0.45 + 0.1\tau$ $\leq 0.55$ | $2.5 + 0.3\tau$ $\leq 2.8$ | $0.6 + 0.1\tau$ $\leq 0.8$ | 1.4 | 1.44 | 0.0066 | 0.9 |





The microphysical properties in Table 1 were used to calculate the optical properties required by the RTM used in iAERUS-GEO (i.e., $P(\xi)$, $\tau$, $\omega$) to perform calculations of TOL reflectance. Calculations were done with the MOPSMAP (Modelled optical properties of ensembles of aerosol particles) software (Gasteiger and Wiegner, 2018) taking into account the SEVIRI spectral responses and results were stored in look-up tables. A Mie code was used for spherical particles whereas the T-matrix code by Mishchenko and Travis (1998) was used for spheroids assuming the aspect ratio distribution described by Dubovik et al. (2006). For dynamic models, calculations were done for values of AOD between 0 and 3 by steps of 0.01.

### 2.6.3 Monthly geographic distribution of aerosol models

The selection of the appropriate optical properties for the processing of each SEVIRI pixel is done based on maps giving the spatial distribution of the available aerosol models. This strategy is borrowed from the MODIS MAIAC C6 algorithm but includes some adjustments such as the extension of the maps to oceans. Another adaptation is the monthly variation of the geographic distribution of Models 6 and 7 to account for the seasonal variations of desert dust and biomass burning smoke. This temporal variation is derived from the CAMS-based monthly maps of AOD. The strategy used in MAIAC for dust and smoke could not be adopted here, as it includes tests based on MODIS channels that are unavailable on SEVIRI. The map for the month of May is shown in Fig. 7 and the approach to construct it is explained in Appendix C. Borders among different models are treated here to avoid spurious AOD boundaries as those referred to in Lyapustin et al. (2018). Buffer zones were introduced along borders in which optical properties of aerosol models are mixed linearly. Albeit being artificial, this approach helps avoiding visual AOD boundaries.

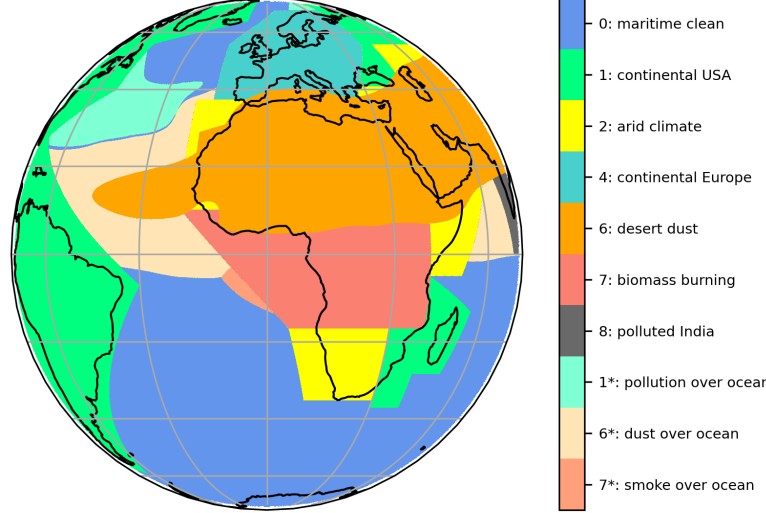

**Figure 7.** Geographic distribution of aerosol models for May. Models 1*, 6*, and 7* correspond to the use of Models 1, 6, and 7 over ocean.



## 3 Validation protocol and data

### 3.1 Experimental design

Experiments were conducted to assess the performances of iAERUS-GEO applied to SEVIRI data. The time period from
January 2012 to September 2013 was considered. Following Ceamanos et al. (2021), the processing of the first three months
was used as spin up time to allow the Kalman filter-based method to provide reliable estimates of surface BRDF. Hence, the
results reported in Sect. 4 correspond to the evaluation of the AOD retrieved by iAERUS-GEO during the 18 months spanning
from April 2012 to September 2013. First, AOD estimates were evaluated with ground observations from the AERONET
network to assess their accuracy and limitations. Second, the satellite aerosol product GRASP/POLDER (Generalized Retrieval
of Atmosphere and Surface Properties/POLarization and Directionality of the Earth's Reflectances) was added to the
comparison to assess the potential of GEO satellites for aerosol remote sensing with respect to LEO spacecraft. Finally, the
period spanning from March to July 2016 was also processed to illustrate the capability of iAERUS-GEO to perform high
temporal resolution monitoring of aerosols. The satellite product MODIS/Dark Target-Deep Blue was used here for
comparison. Details on the input data, the products used for evaluation, and their preprocessing are given below.

### 3.2 Input data

The following inputs were used in this work for the processing of SEVIRI data:

1. Cloud masks were computed from SEVIRI data with the software from the EUMETSAT Satellite Application Facility
on Support to Nowcasting and Very Short Range Forecasting (http://www.nwcsaf.org/). Detection of clouds is done
with the current version of the algorithm that was originally proposed by Derrien and Le Gleau (2005).

2. Meteorological parameters (i.e., total column water vapor, total column ozone, surface pressure, wind speed, and
wind direction) were obtained from ERA5 reanalyses of the European Centre for Medium-Range Weather Forecasts
(Hersbach et al., 2020). The original 3-h data were temporally interpolated to match the 15-min frequency of SEVIRI.

Following Ceamanos et al. (2021), SEVIRI radiances were recalibrated to account for the systematic biases found by Meirink
et al. (2013) with respect to collocated near-nadir reflectance measurements from MODIS. Furthermore, SEVIRI
measurements with a value of SZA or VZA higher than 75° were discarded to mitigate the lower accuracy of the RTM used
in iAERUS-GEO at these geometries and the neglect of the sphericity of the Earth (Korkin et al., 2020). Finally, the greater
difficulty of retrieving AOD in the presence of intense sun glint is solved by not processing ocean measurements with a sun
glint angle (γ) lower than 35°, where $\gamma = \arccos(\mu_s \mu_v - \sin\theta_s \sin\theta_v \cos\varphi)$.

### 3.3 Evaluation data

### 3.3.1 AERONET

AERONET is a network of autonomously operated Sun-sky photometers scattered around the world that provide column-
integrated aerosol properties every few minutes (Holben et al., 1998). All AERONET sites providing valid AOD data from



April 2012 to September 2013 were used here (Fig. 8). AOD observations at 675 nm were resampled to provide collocated 15-min averaged values centered at minutes 00, 15, 30, and 45 to match the SEVIRI acquisition times. Spectral conversion to 635
nm was done to match the SEVIRI VIS06 central wavelength with the Ångström exponent calculated based on the AERONET AOD at 440 and 675 nm. Spatial collocation was done by assigning each ground site to the closest SEVIRI pixel (i.e., no spatial averaging of satellite data over a larger area). AERONET data used in this work correspond to the Version 3 algorithm (Giles et al., 2019) and the quality Level 2.0, including automatic cloud-clearing and pre- or post-field calibration.

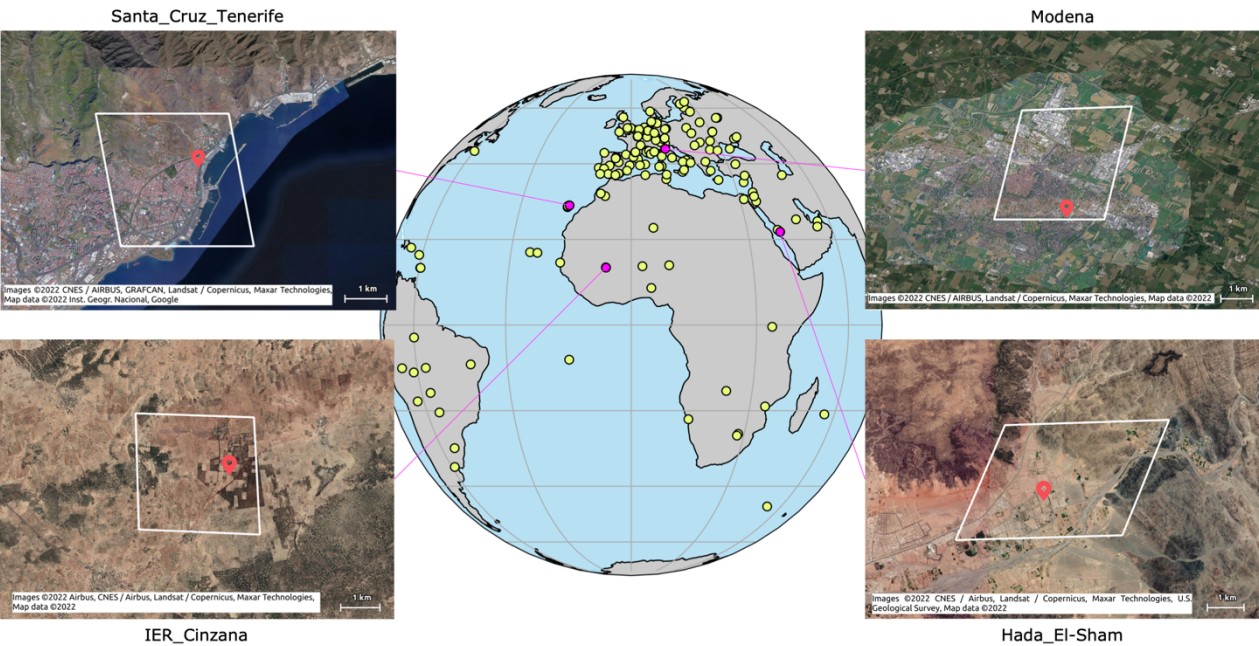

**Figure 8.** Location of AERONET sites used for the assessment of iAERUS-GEO. Sites for which times series are plotted in Sect. 4.4 are highlighted in pink color. The aerial view corresponding to these four sites is also shown, along with the location of the sunphotometer (red pin) and the approximate footprint of the corresponding SEVIRI pixel (white rhombus).

### 3.3.2 GRASP/POLDER

The GRASP/POLDER product provides state-of-the-art satellite observations of AOD from the combination of POLDER
measurements and the retrieval algorithm GRASP. POLDER was a multi-polarization, multi-angular, and multi-spectral imager on the PARASOL LEO satellite which was operative from 2005 to 2013. GRASP performs optimal retrieval of aerosol-surface properties from multi-dimensional remote sensing observations such as POLDER's (Dubovik et al., 2011). The GRASP/POLDER data (hereafter simply referred to GRASP) that were used in this study correspond to the version "Models", which was found to provide the best estimates of total AOD with respect to other versions of GRASP and other satellite
products (Chen et al., 2020). For this study we downloaded data corresponding to Version 2.1 and quality-assured Level-2 (available over a sinusoidal projection at approximately 6 km of resolution) for years 2012 and 2013 from https://www.grasp-





open.com/. GRASP AOD estimates at 670 nm were interpolated to 635 nm with the Ångström coefficient computed based on the GRASP AOD at 565 nm and 670 nm. The resulting data were collocated in time and space for the comparison to AERONET and iAERUS-GEO. First, GRASP AOD estimates were assigned to the closest 15-min interval (centered at minutes

00, 15, 30, or 45) with an existing AERONET estimate within a 1 hour-window centered at the POLDER acquisition time. Second, a nearest neighbor approach was used to select the GRASP pixel corresponding to each AERONET site. This was done to perform a pixel-wise comparison between GRASP and iAERUS-GEO retrievals, as the latter data are at a resolution of ~5-7 km over most of AERONET sites used in this work. According to Chen et al. (2020), the average AOD of the box with size up to $9 \times 9$ pixels was used for some coastal and island sites for which GRASP retrievals were found to be missing.

**3.3.3 DT-DB/MODIS**

A Level-3 daily global aerosol data set based on the combination of products MOD08_D3_v6.1 and MYD08_D3_v6.1 was also used in this work. We used the variable named *AOD_550_Dark_Target_Deep_Blue_Combined* provided by the NASA Earth Observations (https://neo.sci.gsfc.nasa.gov/) at a resolution of 0.1°. This data set results from the combination of the Dark Target (DT) and Deep Blue (DB) algorithms to provide state-of-the-art AOD retrievals from MODIS for a maximized

spatial coverage (Hsu et al., 2013; Levy et al., 2013). The combined DT and DB product (hereafter referred to as DT-DB) provides total AOD at 550 nm from Terra and Aqua, with an overpass time around 10:30 a.m. and 1:30 p.m., respectively.

## 4 Results

### 4.1 Evaluation with AERONET

Figure 9 outlines the evaluation of AOD retrievals from iAERUS-GEO with AERONET data. All satellite estimates from

April 2012 to September 2013 were considered here, with no filtering based on the confidence measure provided by iAERUS-GEO. Figure 9a shows the 2D histogram resulting from the comparison of the two data sets over the 151 available ground sites. Average scores were found to be satisfactory, with a correlation coefficient (R) of 0.77, a mean bias error (MBE) of 0.02, and a root mean squared error (RMSE) of 0.11. The slight overestimation of AOD is caused by residual cloud contamination and a frequent positive bias found over bright surfaces in Northern Africa and the Arabian Peninsula. This can be observed in

Figs. 9b and 9c showing the spatial distribution of the average RMSE and MBE across the AERONET stations. The well-known lower sensitivity to aerosols of satellite data acquired over bright surfaces and the naturally higher values of AOD found in desert regions due to dust activity are the main reasons behind this higher bias. Nevertheless, it is important to notice in Fig. 9d the notably high values of R across the SEVIRI disk including desert sites, which demonstrate the ability of iAERUS-GEO to monitor AOD variations with time even over bright regions. Finally, RMSE and MBE were found to be low for most

sites in Europe, South America, and South Africa.





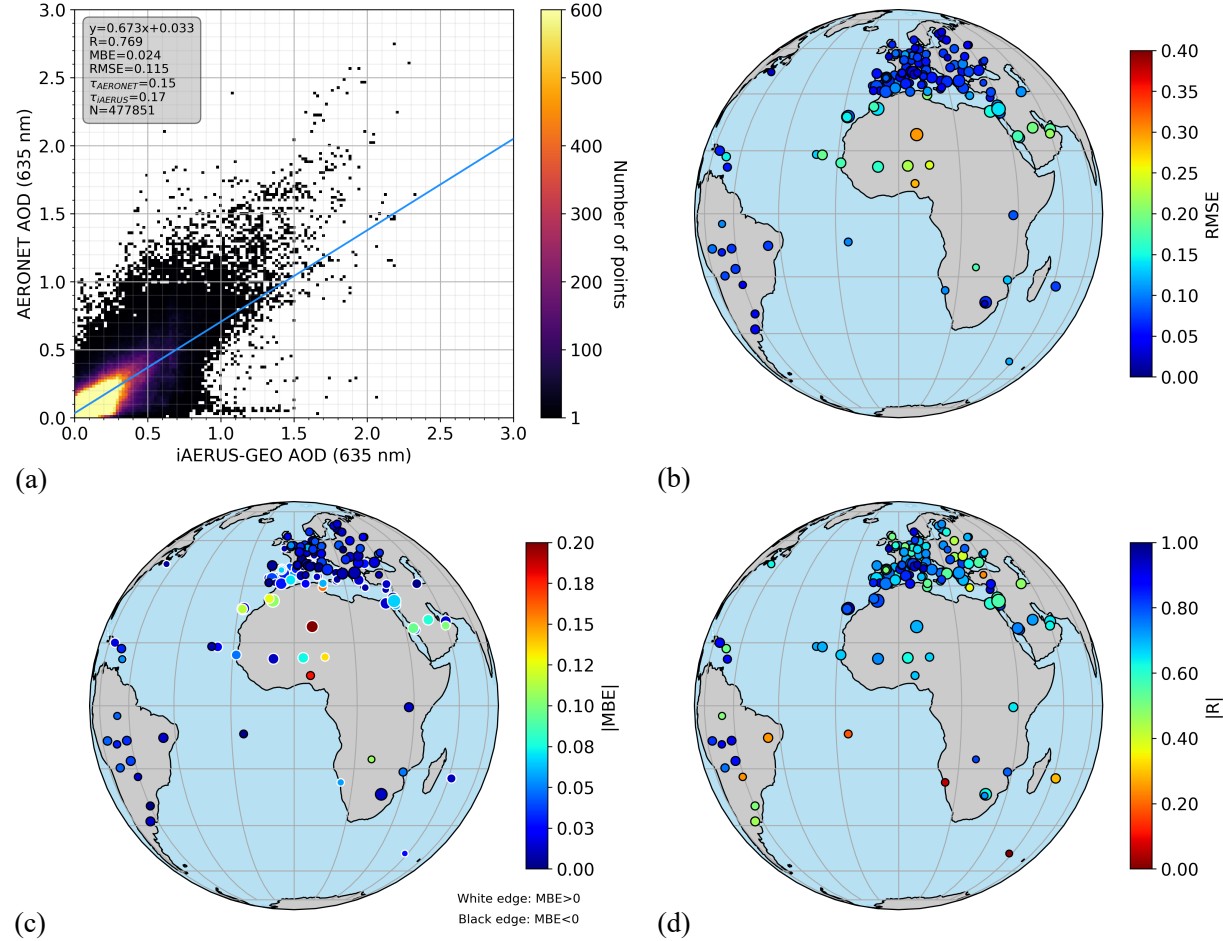

**Figure 9.** Assessment of iAERUS-GEO AOD retrievals from April 2012 to September 2013 with AERONET data. Comparison between the two data sets is illustrated by (a) a 2D histogram and maps showing the average (b) RMSE, (c) MBE, and (d) R obtained for each AERONET site. The size of dots in (b), (c), and (d) is proportional to the number of AOD retrievals.

Figure 10 further investigates the accuracy of iAERUS-GEO with respect to AERONET. First, Fig. 10a confirms the increase of MBE with surface reflectance, with values becoming slightly positive when surface reflectance is greater than 0.2. Second, Fig. 10b shows an increase of AOD bias for high scattering angles, which is related to the decrease in AOD sensitivity of satellite measurements in the backward direction due to the lower aerosol scattering and the increase in surface brightness. The low information content in this case results in AOD overestimation as it can be noticed from the positive slope of the regression

line. Third, the impact of AOD sensitivity on the accuracy of retrievals is investigated in Figs. 10c and 10d by the AOD Jacobian ($K$). Fig. 10c shows how AOD bias increases when $K$ becomes close to zero whereas it remains low for high absolute values of $K$. Fig. 10d shows how bright surfaces are generally behind the lowest values of $K$, which result in higher biases due to the lower sensitivity of satellite measurements to AOD in this case.







**Figure 10.** 2D histograms showing the AOD bias between iAERUS-GEO and AERONET as a function of (a) scattering angle, (b) surface reflectance, and (c) AOD Jacobian. The dependence of (c) on surface reflectance is shown in (d) with a color scatter plot.

### 4.2 Correlation between confidence measure and AOD bias

The relevance of the confidence measure (CM) provided by iAERUS-GEO (Sect. 2.5.3) as proxy for the accuracy of the retrieved AOD is evaluated in Fig. 11. The variation of the average scores obtained from the comparison to AERONET is plotted for an increasing filtering of the AOD estimates based on their value of CM. For example, the case referred to as "CM>2" corresponds to the filtering of estimates corresponding to values of CM=1 and CM=2. Overall, Fig. 11 shows a steady improvement of the accuracy of iAERUS-GEO estimates, with a significant improvement in terms of R, RMSE, and MBE. The punctual decrease of R for the cases "CM>3" and "CM>5" raises from the filtering of AOD estimates corresponding to





bright surfaces, which are often linked to high R values due to the strong variation of aerosol load over deserts. An increasing

loss in the number of retrievals (N) is also observed, which becomes especially remarkable from the case "CM>4" on.

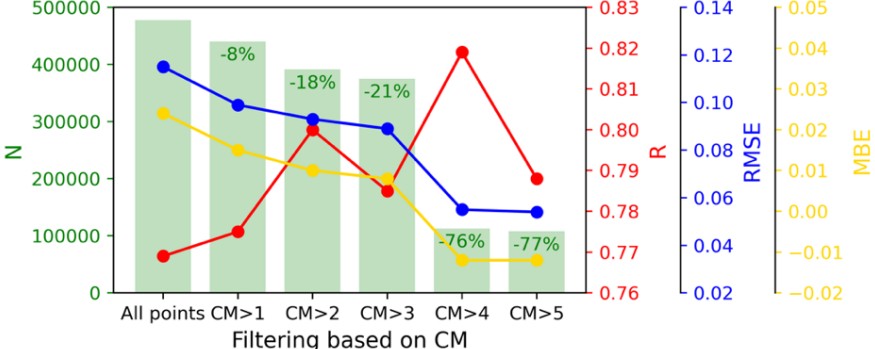

**Figure 11.** Variation of the averages scores (R, RMSE, MBE) obtained from the comparison to AERONET and the number of retrievals (N) according to an increasing filtering of iAERUS-GEO AOD retrievals based on the confidence measure (CM).

We consider the case "CM>2" to be optimal due to its significantly improved scores (i.e., R=0.800, RMSE=0.093,

MBE=0.010) and its moderate decrease in N of 18% with respect to the case without filtering. Figure 12 further investigates

the accuracy of iAERUS-GEO retrievals in this case. First, Fig. 12a shows that the increase in R mostly comes from AERONET

sites located across the dust belt and in Southern Europe. As discussed before, these regions are generally related to brighter

surfaces which result in a lower sensitivity to aerosol load, thus a lower AOD Jacobian and a lower CM. Figure 12b shows

how the decrease in number of retrievals is more significant over the same stations, with values ranging from less than 20%

for sites in Spain to more than 70% in a few sites in Africa. Finally, Figs. 12c and 12d show the 2D histograms of AOD bias

as a function of surface reflectance and scattering angle for the case "CM>2" (to be compared with Figs. 10a and 10b

considering all retrievals). A notable bias reduction and a significant decrease of the regression slopes are observed, thus

corroborating the ability of the CM to filter the overestimated AOD values.





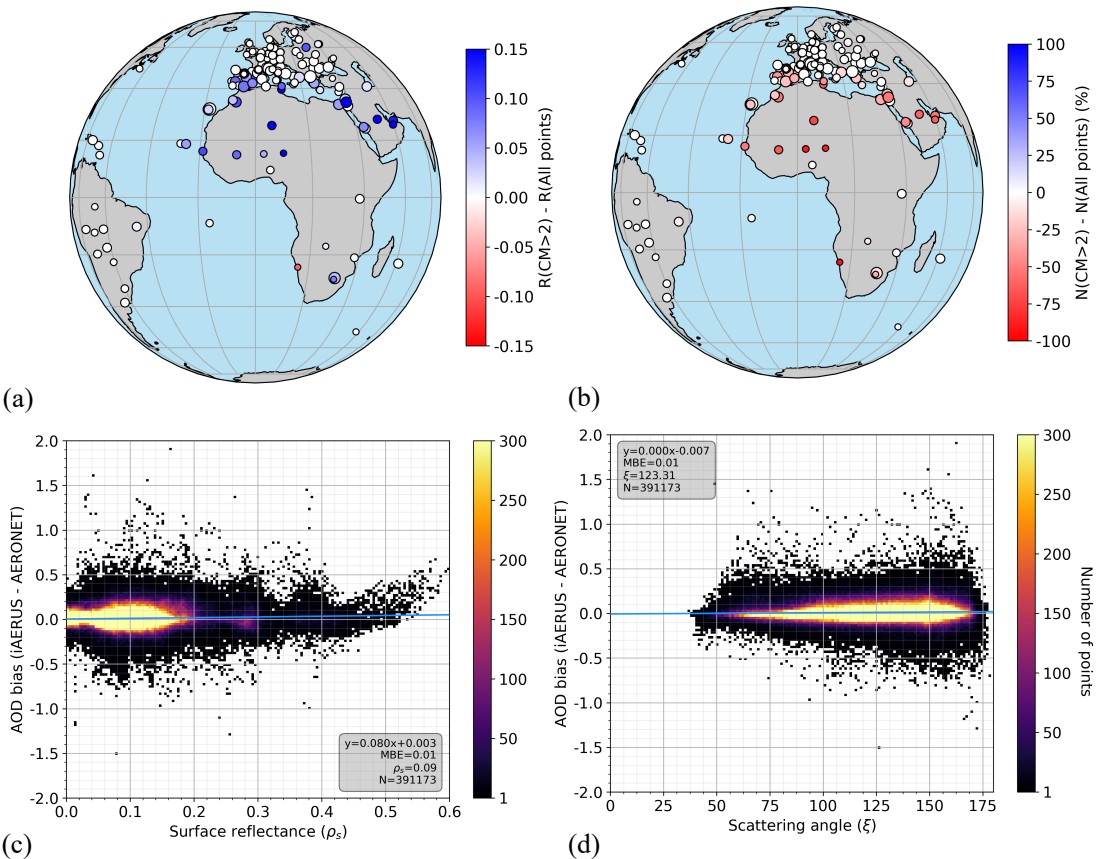

**Figure 12.** Evaluation of the AOD retrievals for the case "CM>2". Score maps in (a) and (b) show the difference in R and N, with respect to the case with no filtering. 2D histograms in (c) and (d) show the relation between AOD bias and surface reflectance and scattering angle.

## 4.3 Comparison to GRASP/POLDER

The quality of iAERUS-GEO is further assessed with the satellite product GRASP. Only retrievals with CM>2 were considered here according to the results reported in the previous section. First, Figs. 13a and 13b summarize the comparison of the two satellite AOD products to AERONET collocated data from April 2012 to September 2013. Overall, both data sets were found to provide similar scores, with a slightly lower error for iAERUS-GEO (e.g., RMSE of 0.093 against 0.102) and higher R for GRASP (0.885 against 0.800). The biggest difference lies in the number of estimates, which is remarkably larger for iAERUS-GEO (391173 versus 7090, i.e., 55 times more) due to the higher number of measurements made available by GEO satellites with respect to LEO missions. Figs. 13c and 13d illustrate the similar spatial distribution of the average RMSE for the two satellite products, although some differences can be observed. On the one hand, iAERUS-GEO was found to provide larger errors over bright surfaces due to the limited information from SEVIRI over this type of regions as it was previously discussed. On the other hand, GRASP shows larger errors for some sites in Northern Europe probably due to a degraded characterization of surface reflectance caused by the reduced chances of getting cloud-free observations from POLDER in winter.



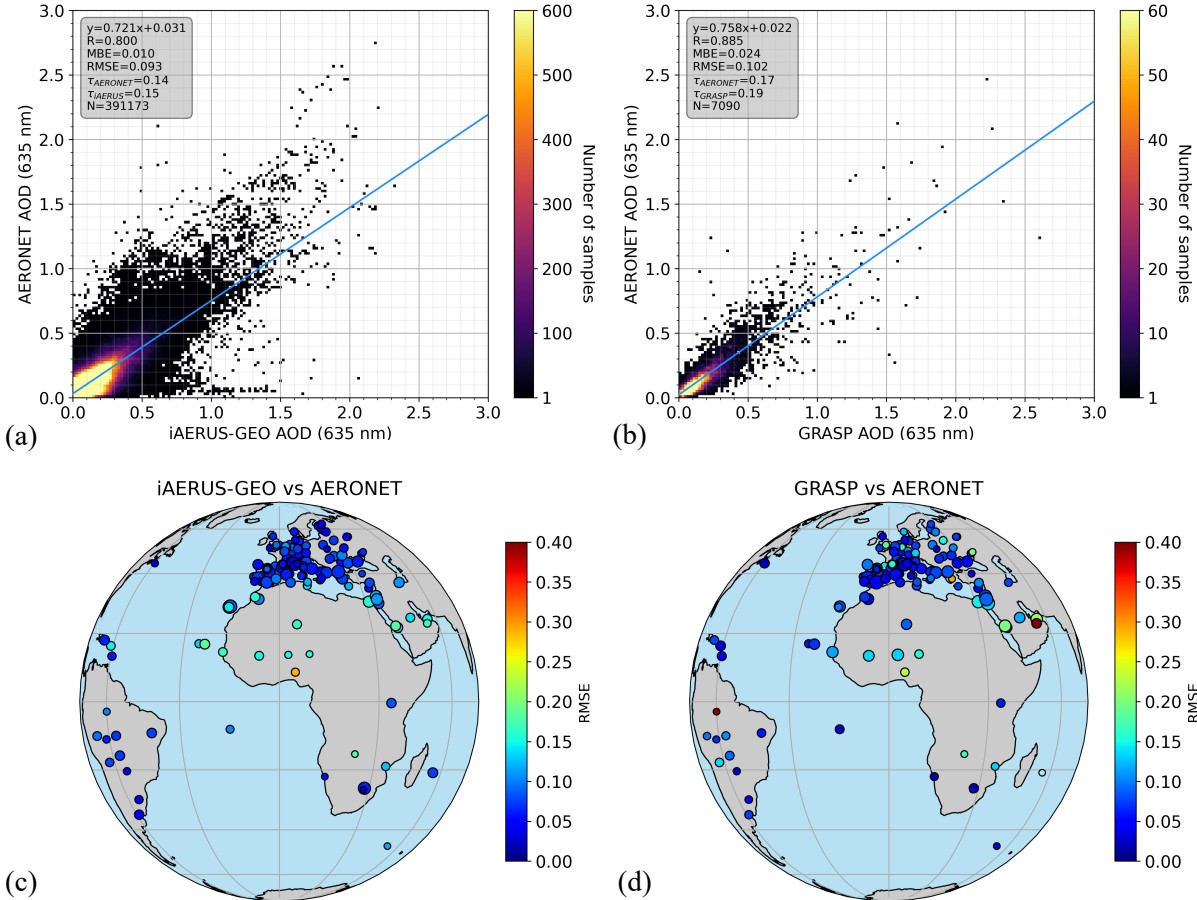

**Figure 13.** Assessment of two satellite AOD data sets with AERONET from April 2012 to September 2013. Comparison between iAERUS-GEO (CM>2) and AERONET is shown by (a) a 2D histogram and (c) a map showing the average RMSE per station. The same plots are given for the comparison between GRASP and AERONET in (b) and (d). Note the different color bar scale in (a) and (b).

Second, Fig. 14 summarizes the direct comparison between iAERUS-GEO and GRASP. The collocation of the two data sets for this exercise resulted in the selection of iAERUS-GEO data corresponding to the overpass time of PARASOL only (which went from 2:45 p.m. in April 2012 to 4:00 p.m. in September 2013 at the equator due the satellite drift). Figure 14a shows a notable agreement between the two satellite products, which is however slightly less satisfactory than the individual comparisons to AERONET especially in terms of R. The reason behind this result is investigated in Fig. 14b, which shows a clear West-East gradient of the average correlation between the two data sets. The poorer agreement observed in the East was found to be linked to a decreased quality of iAERUS-GEO over these regions due to a lower information content of SEVIRI data at the PARASOL overpass time caused by the occurrence of high scattering angles (Fig. 14c). On the other hand, western regions were observed by SEVIRI at lower scattering angles, which allowed a more reliable retrieval thanks to the higher sensitivity to AOD (Fig. 12d). The variation of scattering angle across the MSG disk comes from the fact that the PARASOL overpass happens at the same local time but at different UTC time (i.e., morning in the East and afternoon in the West).





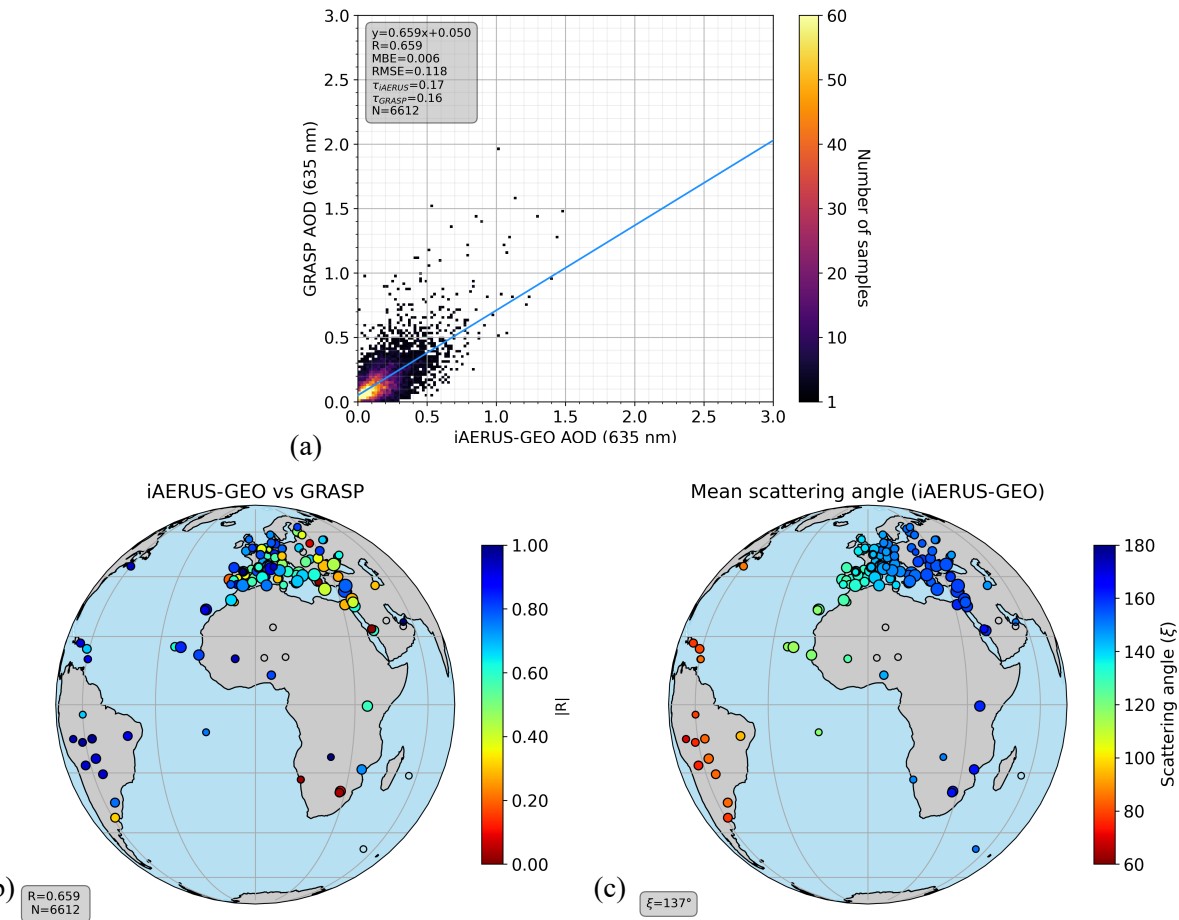

**Figure 14.** Comparison between iAERUS-GEO (CM>2) and GRASP by (a) a 2D histogram and (b) a map giving the average R for each AERONET site. The average scattering angle of the iAERUS-GEO retrievals considered in this comparison is given in (c).

### 4.4 Diurnal variation of retrieved AOD

The high frequency aerosol observations that are possible thanks to GEO remote sensing are illustrated in Fig. 15. Time series of AOD provided by iAERUS-GEO (CM>2) and AERONET are shown for four ground sites (Fig. 8 for their location and aerial view) and 1-month time periods that were selected according to land cover type and aerosol activity. The first station *Santa_Cruz_Tenerife*, situated along the coast of Tenerife island, is frequently reached by mineral dust transported from the Sahara Desert. Figure 15a shows the ability of iAERUS-GEO to monitor the daily and diurnal variation of AOD during summer 2012 —with three consecutive dust events— resulting in high R values with respect to AERONET. Some cloud contamination probably due to a faulty cloud mask can be seen on August 7th and 17th. Figure 15b corresponds to the site *Modena* located in the middle of the Po Valley, the most industrialized area in Italy. Although aerosol load is relatively stable throughout the period of study in 2013, the few daily (e.g., pollution peak on June 12th) and diurnal (e.g., increase of AOD on June 22nd and 23rd) variations that can be observed are well captured by iAERUS-GEO in most occasions. The third station *Hada_El-Sham*





is situated in a background area about 65 km east of the city of Jeddah. Figure 15c shows the agreement between AERONET and iAERUS-GEO for this site, which experienced a rapid evolution of aerosol load during spring 2013, with values fluctuating

between 0 and 1.5 due to recurrent dust transport from the near deserts in Saudi Arabia. The higher difficulty for aerosol retrieval over this barren site with a rather bright surface is seen in the missing iAERUS-GEO retrievals in the local afternoon resulting from the CM-based filtering. The decrease in information content of SEVIRI at that time of the day can also be seen by looking at the spurious diurnal cycle on May 15th, for example, for which aerosols were almost undetectable due to the low AOD and the unfavorable measuring scattering angle. Finally, the fourth station *IER_Cinzana* located in the heart of the Sahel

was selected due to the high difficulty of retrieving AOD over this rural area, with a rather high surface reflectance and under the influence of several aerosol types. The low sensitivity to AOD over this site can be noticed from the lower number of iAERUS-GEO retrievals with respect to previous sites. However, the CM-based filtering behind this fact proves to work well, as scores are satisfactory with regard to AERONET. Figure 15 also shows how GRASP retrievals (in orange color) correlate well with AERONET, although they are available at a much lower temporal frequency compared to iAERUS-GEO.

**Figure 15.** Time series of AOD from iAERUS-GEO (CM>2; in blue color), AERONET (in black color), and GRASP (in orange color) for sites (a) *Santa_Cruz_Tenerife*, (b) *Modena*, (c) *Hada_El-Sham*, and (d) *IER_Cinzana* during 1-month time periods in 2012 and 2013.



### 4.5 High temporal resolution monitoring of AOD during a dust transport event in Southwestern Europe

The potential of iAERUS-GEO and SEVIRI/MSG for aerosol retrieval at high frequency is further illustrated by focusing on
an episode of dust transport in July 2016. At that time, large quantities of mineral dust uplifted from the Sahara Desert were
transported into the Atlantic Ocean and the Mediterranean basin. Figure 16a shows a color composite of the SEVIRI radiance
images acquired during this event, at 10:30 a.m. on July 20th, 2016. The map of AOD retrieved by iAERUS-GEO on the same
date and time is seen in Fig. 16b and shows a massive aerosol plume spanning from Eastern Caribbean to Spain. The gray
areas correspond to cloudy regions that were not processed and, to a lesser extent, to filtered retrievals (only estimates with
CM>2 were considered here) and not processed pixels (e.g., Western South America due to the high solar geometries during
sunrise). The reliability of this AOD map is confirmed by its similarity to Fig. 16c, which corresponds to the AOD provided
by the DT-DB algorithm applied to MODIS-Terra on the same date. Some differences can also be noticed including a higher
data completeness from iAERUS-GEO in some regions (e.g., Atlantic Ocean) except for bright land areas (e.g., North Africa
and the Arabian Peninsula) for which the spectral information content of SEVIRI is lower with respect to MODIS. Finally, it
is important to stress that the coverage of iAERUS-GEO becomes much higher at the end of the day with respect to DT-DB
due to the existence of one AOD map every 15 minutes during daytime.

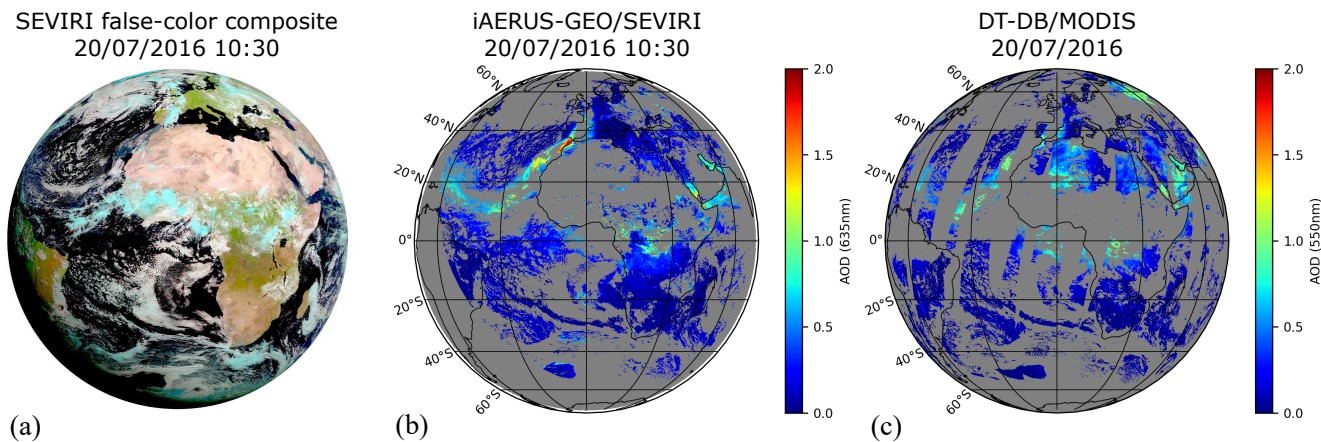

**Figure 16.** (a) Color composite of SEVIRI radiance images acquired at 10:30 a.m. on July 20th, 2016. (b) Map of AOD retrieved by iAERUS-GEO from SEVIRI on the same date and time. (c) Map of AOD retrieved by DT-DB from MODIS-Terra on the same date.

Figure 17 zooms in over Southwestern Europe from July 19th to July 22nd, 2016, and shows how mineral dust was carried into
the Mediterranean basin through the Strait of Gibraltar. The high temporal resolution of iAERUS-GEO allows the fine temporal
monitoring of these aerosol particles, which reached the Balearic Islands in the morning of July 20th, Central Italy in the
afternoon of July 21st, and Sicily in the morning of July 22nd.





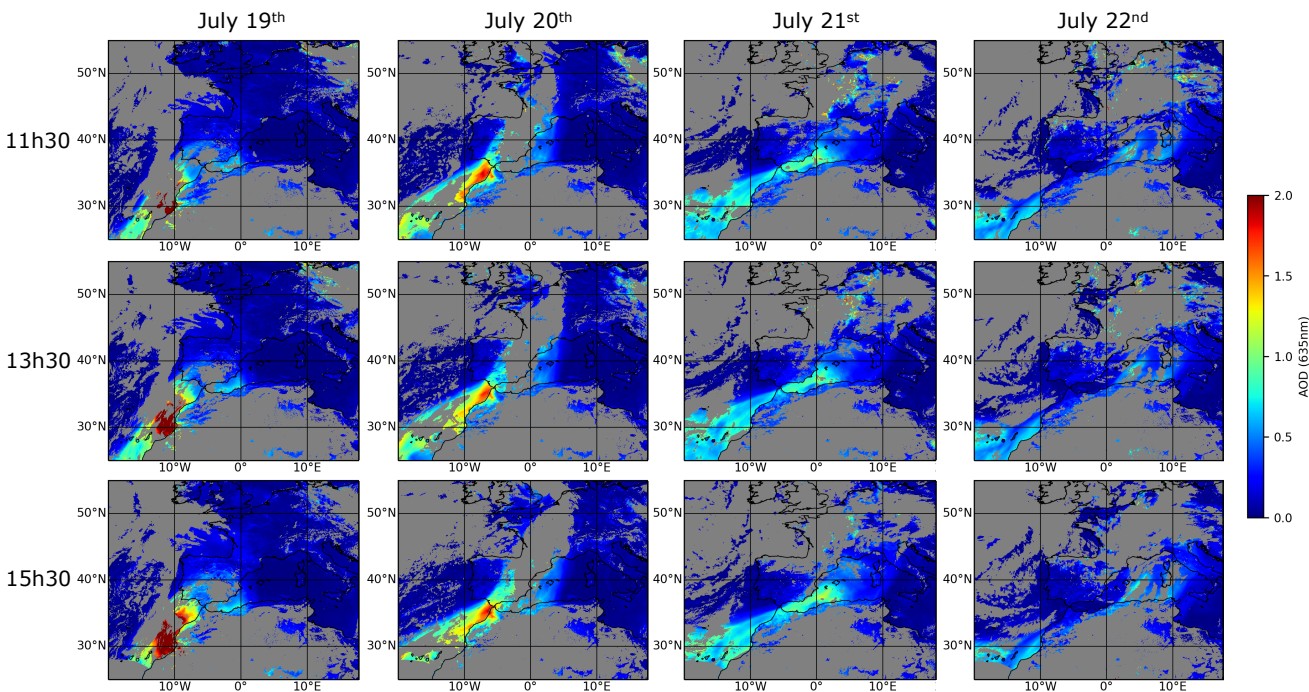


**Figure 17.** AOD from iAERUS-GEO at (top) 11:30 a.m., (middle) 1:30 p.m., and (bottom) 3:30 p.m. for four consecutive days in July 2016.

The accuracy of iAERUS-GEO during this dust event was assessed with AERONET. Figure 18 (left) shows the ground sites that were selected according to their location along the dust transportation. Figure 18 (right) shows the average scores obtained from the comparison between satellite and ground AOD data from July 19th to July 21st. As it can be seen, iAERUS-GEO was

found to provide reliable results with R between 0.82 and 0.98, RMSE between 0.03 and 0.12, and absolute MBE lower than 0.05. The number of AOD estimates reached a value of 1023, resulting in an average of 28 retrievals per day and per site.

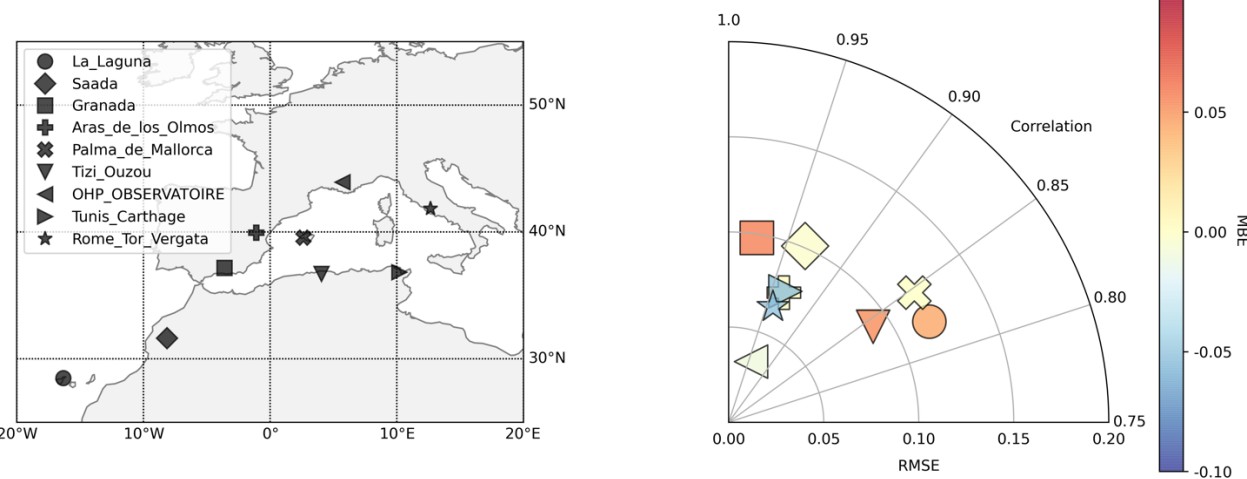

**Figure 18.** Assessment of iAERUS-GEO with AERONET during the dust event in July 2016. (left) Selected sites. (right) Average scores.



## 5 Conclusions and future work

This article describes a new algorithm iAERUS-GEO (instantaneous Aerosol and surfacE Retrieval Using Satellites in GEOstationary orbit) that performs high frequency monitoring of atmospheric aerosols over land and ocean from geostationary meteorological satellites. Here we applied this method to data from the SEVIRI imager on the satellite MSG, resulting in the estimation of a map of total AOD every 15 minutes during daytime. Aerosol load is estimated from the "red" channel VIS06 centered at 635 nm, which corresponds to the shortest measuring wavelength of SEVIRI. Extensive assessment with collocated

AERONET ground observations highlighted the satisfactory quality of iAERUS-GEO estimates (i.e., R=0.80, RMSE=0.093, MBE=0.01). This accuracy was found to be comparable to the state-of-the-art satellite aerosol product GRASP derived from the POLDER sensor. However, only iAERUS-GEO was found to be able to detect the fine temporal variations of aerosol load during the day. This is possible thanks to the larger number of retrievals (55 times more for iAERUS-GEO compared to GRASP) coming from the higher acquisition frequency of GEO missions with respect to LEO sensors such as POLDER.

This study also shows that the accuracy of iAERUS-GEO may vary during the day due to the changing information content on aerosols of GEO measurements. This change in the sensitivity to AOD is caused by the broad range of scattering angles covered during the day due to the motion of the Sun with respect to the satellite. The difficulty to estimate AOD in the occurrence of low information content is aggravated for barren regions due to their higher brightness. In this case, however, the confidence measure provided by iAERUS-GEO was proven effective to filter most of potentially biased AOD retrievals.

The instantaneous estimation of AOD presented in this work will certainly improve with the upcoming Meteosat Third Generation-Imager (MTG-I) satellites from EUMETSAT (Holmlund et al., 2021). This next generation of geostationary meteorological satellites will be equipped with the Flexible Combined Imager (FCI), which will outperform SEVIRI in terms of spatial, spectral, and temporal characteristics. In particular, retrieval over bright surfaces is expected to be eased thanks to FCI additional channels in the "blue" and "green" wavelengths, which are already used in other aerosol algorithms to exploit

the fact that surfaces are generally darker at shorter wavelengths (e.g., Hsu et al., 2013). The enhanced spectral sensitivity of FCI —with 8 channels in the visible and near infrared range instead of 3 for SEVIRI— could also be useful to distinguish among different types of aerosols from space as it is done by some algorithms processing multiple spectral channels simultaneously (e.g., Lyapustin et al., 2018). This information on the particle type would help to improve the selection of aerosol models in iAERUS-GEO currently based on a monthly climatology. The restricted domain of validity of the current

RTM —including SMAC and the MSA model— and other limitations including the neglect of the Earth's sphericity will also be improved in the future adaptation of iAERUS-GEO to FCI/MTG-I.

In conclusion, this work proves the ability of geostationary satellites to break the temporal barrier of aerosol observations and opens the door to scientific studies that are currently unachievable with LEO missions. The proposed method iAERUS-GEO was successfully tested on SEVIRI/MSG data but could be applied to other geostationary meteorological missions and to a

constellation of this type of satellites —as it was done in Ceamanos et al., (2021)— to achieve an unprecedented global monitoring of aerosols at high frequency.





**Appendix A: Radiative transfer modeling**

**A.1 Modified Sobolev's Approximation (MSA) for aerosol contribution**

MSA (Katsev et al., 2010) uses approximate analytical functions for aerosol radiative terms by adopting the solution of the truncation of the phase function. The removal of the elongated forward scattering peak of aerosols does not practically affect the aerosol signal in the backward hemisphere, which is the angle range in which most of spaceborne satellites operate.

**A.1.1 Truncation of phase function**

Let $P(\xi)$, $\tau$, $\omega$, $g$ be the optical properties describing a specific aerosol layer. The scattering phase function $P(\xi)$ can be truncated according to the well-known Delta-Eddington approximation (Joseph et al., 1976)

$$\tilde{P}(\xi) = \frac{P(\xi)}{1-\eta} \text{ if } \xi > \xi^*$$

$$\tilde{P}(\xi) = 0 \text{ if } \xi < \xi^*, \tag{A1}$$

where $\xi^*$ is the truncating scattering angle and $\eta$ is the truncated area of the phase function calculated as

$$\eta = \frac{1}{2}\int_0^{\xi^*} P(\xi) \sin \xi \, d\xi. \tag{A2}$$

Using this technique, the original aerosol medium is transformed into a new medium defined by the tilde variables $\tilde{P}(\xi)$, $\tilde{\tau}$, $\tilde{\omega}$, $\tilde{g}$ that are calculated with Eq. A1 and the following expressions

$$\tilde{\tau} = (1 - \omega\eta)\tau, \tag{A3}$$

$$\tilde{\omega} = \frac{1-\eta}{1-\omega\eta}\omega, \tag{A4}$$

$$\tilde{g} = \frac{\int_{\xi^*}^{\pi} \tilde{P}(\xi) \cos \xi \sin \xi \, d\xi}{\int_{\xi^*}^{\pi} \tilde{P}(\xi) \sin \xi \, d\xi}. \tag{A5}$$

In iAERUS-GEO, all the radiative transfer calculations are made using the tilde variables and, only at the end of the inversion, $\tilde{\tau}$ is converted to $\tau$ with Eq. A3.

**A.1.2 Reflectance**

Following the Sobolev's approximation (Sobolev, 1975), and $\mu_s$ and $\mu_v$ being the cosines of the solar and view zenith angles, the reflectance of the new aerosol medium can be represented as the sum of single reflection and multiple reflection

$$\rho_{aer}(\mu_s, \mu_v, \varphi) = \rho_{aer}^{SS}(\mu_s, \mu_v, \varphi) + \rho_{aer}^{MS}(\mu_s, \mu_v), \tag{A6}$$

with the single scattering term being





$$\rho_{aer}^{SS}(\mu_s, \mu_v, \varphi) = \widetilde{\omega}\widetilde{P}(\xi)\rho_1, \tag{A7}$$

with

$$\rho_1 = \frac{1}{4(\mu_s + \mu_v)}(1 - e^{-\tilde{\tau}m}), \tag{A8}$$

with $m$ being the airmass ($m = \mu_s^{-1} + \mu_v^{-1}$) and the multiple scattering term being

$$\rho_{aer}^{MS}(\mu_s, \mu_v) = 1 - \frac{R(\tilde{\tau}, \mu_s)R(\tilde{\tau}, \mu_v)}{4 + (3 - \widetilde{x_1})\tilde{\tau}} + [(3 + \widetilde{x_1})\mu_s\mu_v - 2(\mu_s + \mu_v)]\rho_1, \tag{A9}$$

with

$$R(\tilde{\tau}, \mu) = 1 + 1.5\mu + (1 - 1.5\mu)e^{-\tilde{\tau}/\mu}, \tag{A10}$$

where $\widetilde{x_1} = 3\tilde{g}$ is the first coefficient of the expansion of $\tilde{P}(\xi)$ into a series of Legendre polynomials.

All these analytical solutions are only valid for not very elongated phase functions, which emphasizes the importance of the truncation procedure.

### A.1.3 Transmittance and albedo

Following Katsev et al. (2010), the aerosol upwelling and downwelling transmittances read

$$T_{aer}^{\uparrow}(\mu) = \exp\left[-\tilde{\tau}\left(1 - \widetilde{\omega}\widetilde{F_1}\right)/\mu_v\right]$$

$$T_{aer}^{\downarrow}(\mu) = \exp\left[-\tilde{\tau}\left(1 - \widetilde{\omega}\widetilde{F_1}\right)/\mu_s\right], \tag{A11}$$

where

$$\widetilde{F_1} = 1 - \frac{1 - \tilde{g}}{2}, \tag{A12}$$

and the spherical albedo is

$$a_{aer} = \frac{\tilde{\tau}}{\tilde{\tau} + 4/(3 - \widetilde{x_1})}. \tag{A13}$$

### A.2 Kernel-based model for surface contribution

### A.2.1 Bidirectional reflectance for land

Land surface BRDF is decomposed into three kernels following the hotspot-corrected Ross-Li model (Maignan et al., 2004). The first kernel represents the isotropic (Lambertian) scattering

$$f_0^l = 1. \tag{A14}$$





The second kernel models the geometric-optical surface scattering as from scenes containing three-dimensional objects that

cast shadows and are mutually obscured from view at off-nadir angles

$$f_1^l(\theta_s, \theta_v, \varphi) = \frac{m}{\pi}(t - \sin t \cos t - \pi) + \frac{1+\cos\xi}{2\mu_s\mu_v}, \tag{A15}$$

with

$$\cos t = \frac{2}{m}\sqrt{\Delta^2 + (\tan\theta_s \tan\theta_v \sin\varphi)}, \tag{A16}$$

The third kernel models the radiative transfer-type volumetric scattering as from horizontally homogeneous leaf canopies

$$f_2^l(\theta_s, \theta_v, \varphi) = \frac{4}{3\pi}\frac{1}{\mu_s+\mu_v}\left[\left(\xi - \frac{\pi}{2}\right)\cos(\pi - \xi) + \sin(\pi - \xi)\right]\left[1 + \left(1 + \frac{\xi}{\xi_0}\right)^{-1}\right] - \frac{1}{3}, \tag{A17}$$

where the factor $1 + \left(1 + \frac{\xi}{\xi_0}\right)^{-1}$ takes into account the so-called hotspot effect that results in the increase of the surface

reflectance of vegetation as we get close to the backscattering geometries. The parameter $\xi_0$ is a characteristic angle that can

be related to the ratio of scattering element size and the canopy vertical density. A constant value of $\xi_0 = 1.5°$ is adopted

according to Maignan et al., (2004) to avoid the addition of a free parameter in the BRDF modeling.

**A.2.2 Bidirectional reflectance for ocean**

Ocean surface BRDF is decomposed into two kernels according to the absence or not of scattering directionality.

The first kernel models the isotropic reflectance from whitecaps (foam) and underlight (radiance reflected just below the water
surface)

$$f_0^o = 1. \tag{A18}$$

The second kernel models the anisotropic sun glint (specular reflection of rays of light by the sea in the satellite direction),

thus making $k_1^o$ the fraction of surface providing Fresnel's reflection. Function $f_1^o$ depends on solar and view geometry, wind

speed ($w$), and wind direction ($\chi_w$), and is expressed according to the model of Cox and Munk (1954) multiplied by a function

accounting for shadowing from rough sea surfaces ($S$)

$$f_1^o(\theta_s, \theta_v, \varphi, w, \chi_w) = \frac{\pi p(Z_u, Z_v)R_f}{4\mu_s\mu_v\mu_\beta^4}S, \tag{A19}$$

where $p(Z_u, Z_v)$ describes the probability distribution of surface facets

$$p(Z_u, Z_v) = \frac{1}{2\pi\sigma_u\sigma_v}e^{-0.5(Z_u^2/\sigma_u^2 + Z_v^2/\sigma_v^2)}, \tag{A20}$$

with the surface slope defined in a particular coordinate system ($u, v$) defined based on wind direction and with the two

components being $Z_u = Z_x\cos\chi_w + Z_y\sin\chi_w$ and $Z_v = -Z_x\sin\chi_w + Z_y\cos\chi_w$. The components in the original coordinate





system $(x, y)$ are $Z_x = (-\sin\theta_v \sin\varphi)/(\mu_s+\mu_v)$ and $Z_y = (\sin\theta_s +\sin\theta_v \cos\varphi)/(\mu_s+\mu_v)$. Finally, the mean square slopes
components are taken as $\sigma_u^2 = 0.00192w + 0.003$ and $\sigma_v^2 = 0.00316w$ according to Cox and Munk (1954) for a clean sea surface. The facet tilt $(\beta)$ in Eq. A19 is defined as

$$\mu_\beta = \cos\beta = \frac{\mu_s+\mu_v}{\sqrt{2+2\cos 2\Theta}} \tag{A21}$$

where the scattering angle $\Theta$ between the surface facet and the incident beam is $\cos 2\Theta = \mu_s\mu_v + \sin\theta_s \sin\theta_v \cos\varphi$.
The shadowing function $(S)$ and the Fresnel reflection coefficient $(R_f)$ in Eq. A19 are calculated analogously to the GRASP algorithm (Dubovik et al., 2011), which follows the work of Mishchenko and Travis (1997). The calculation of the latter parameter requires the real component of the refractive index of water, which was calculated to be equal to 1.3386 for SEVIRI channel VIS06 at 0.635 $\mu$m by interpolating the spectral values given in Table 3 of Sayer et al. (2010).

### A.2.3 Calculation of albedo

The spherical albedo of the surface in Eq. 1 is calculated as

$$a_s = \iint_{2\pi} \rho_s(\theta_s, \theta_v, \varphi) \cos\theta_v \cos\theta_s \, d\Omega_v d\Omega_s, \tag{A22}$$

where $d\Omega_v = \sin\theta_v \, d\varphi_v$ represents the view hemisphere and $d\Omega_s = \sin\theta_s \, d\varphi_s$ represents the solar hemisphere.

### Appendix B: Modified kernels for daily inversion of surface BRDF

For the $n$ surface terms, kernel functions are modified by doing

$$f_i' = f_i \frac{T_{aer}^\uparrow T_{aer}^\downarrow}{1-a_{aer}a_s}, \tag{B1}$$

with $f_i$ being the original surface kernels, respectively, $f_i^l$ and $f_i^o$ for land and ocean (Appendixes A.2.1 and A.2.2).
The kernel for aerosol single scattering term is derived from Eqs. A7 and A8

$$f_i' = \omega P(\xi) \frac{1}{4(\mu_s+\mu_v)} R_{4/3}, \tag{B2}$$

in which the term $(1 - e^{-\tau m})$ has been approximated by the ratio of two polynomial expansions making $\tau$ a multiplying factor

$$R_{4/3} = \left[\frac{840-60m\tau+20(m\tau)^2-(m\tau)^3}{840-360m\tau+60(m\tau)^2-4(m\tau)^3}\right] m\tau. \tag{B3}$$

This approximation was found to provide a precision of 99,7% for zenith angles up to 70° in Carrer et al., (2010).

### Appendix C: Calculation of monthly geographic distribution of aerosol models

Three steps were followed to construct the maps that indicate the aerosol models to be used in iAERUS-GEO:



1. The geographic distribution used in MODIS MAIAC C6 for land aerosol models (Fig. 4 in Lyapustin et al., 2018) was projected onto the Meteosat disk. The resulting map (Fig C1a) is used as background for each month.


2. Monthly geographic distribution was calculated for Models 6 (desert dust) and 7 (biomass burning) based on the CAMS monthly AOD maps (Sect. 2.6.1) by selecting the pixels satisfying $AOD_{DU} > 0.7 AOD_{total}$ and $AOD_{OM} + AOD_{BC} > 0.7 AOD_{total}$, respectively (Fig. C1b for May). The resulting masks are superposed to the background map.

3. The presence of land-originated anthropogenic pollution, dust, and smoke over ocean is considered by respectively using Models 1, 6, and 7 for ocean pixels without a previously ascribed model and satisfying $AOD_{SU} + AOD_{BC} >$


$0.3 AOD_{total}$, $AOD_{DU} > 0.3 AOD_{total}$, and $AOD_{OM} + AOD_{BC} > 0.3 AOD_{total}$ (Fig. C1c for May). The lower threshold values used here are justified by the less reliable AOD estimates obtained with Model 0 unless pure sea salt aerosols are present. Remaining pixels were assigned to Model 0.

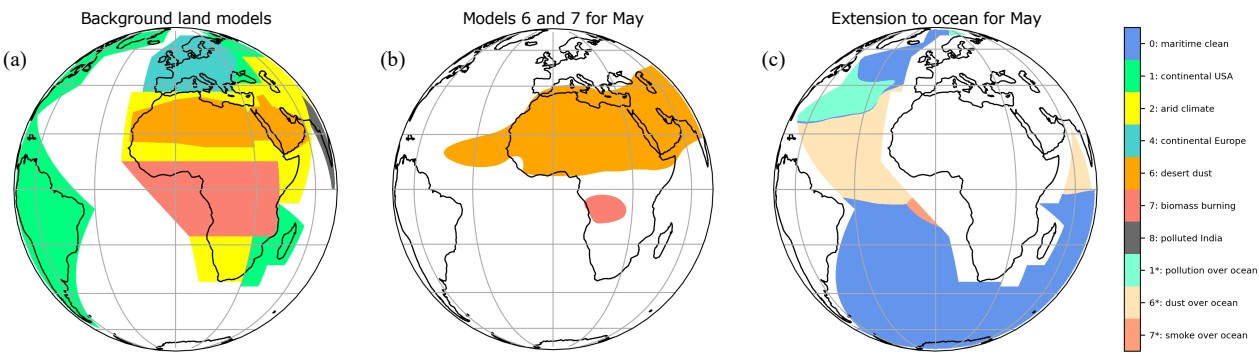

**Figure C1.** Construction of the aerosol model maps for May (Fig. 7) based on (a) the background land models superposed with (b) the
extension of Models 6 and 7 beyond their initial boundaries and (c) the models for the remaining ocean pixels.

*Data availability*. The data used in the article are available upon request from the corresponding author.

*Author contributions*. XC developed the retrieval algorithm with contributions of DC, AG, JG, AL, JLA, and IK. BS, XC, and DC developed the scientific code. XC and BS performed the data processing with the support of JR. XC and SM designed the experiments for validation. XC analyzed the results and prepared the manuscript with contributions from SM, JLA, JR, AG, AL, JG, and IK.

*Competing interests*. The contact author has declared that none of the authors has any competing interests.

*Acknowledgments*. We thank the investigators and their staff for establishing and maintaining the AERONET sites used in this work. The French space agency (CNES) and Jan Fokke Meirink from KNMI are acknowledged for providing the SMAC coefficients and the recalibration coefficients, respectively, that were used to process SEVIRI data. The SATMOS/CMS center is acknowledged for providing the satellite radiances and cloud masks. Abderrahmane Aiche is thanked for its contribution to the generation of the aerosol models. Anthéa
Delmotte is thanked for generating the aerial views of the ground stations. We finally acknowledge all the scientists with whom we had fruitful discussions on the development and validation of iAERUS-GEO, including the GRASP team, Samuel Rémy, Julien Sablon, Alexander Kokhanovsky, and the EUMETSAT Cloud and Aerosol team.

*Financial support*. This research has been partly supported by "Programme national de télédétection spatiale" from INSU-CNRS (PNTS-2021-04) and EUMETSAT (RfQ_18/215735, LSA-SAF_2015-02).



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
