# Peer review of "Instantaneous aerosol and surface retrieval using satellites in geostationary orbit (iAERUS-GEO) – Estimation of 15-min AOD from MSG/SEVIRI and evaluation with reference data"

_Atmospheric Measurement Techniques, 2023_

## Referee Comment (RC2)

**Instantaneous aerosol and surface retrieval using satellites in geostationary orbit (iAERUS-GEO) – Estimation of 15-min AOD from MSG/SEVIRI and evaluation with reference data**

**Review of Ceamanos et al., 2023**

**General**

Ceamanos *et al.* have developed and extended the iAERUS-GEO retrieval algorithm to retrieve Aerosol Optical Depth (AOD) at a very high temporal frequency (every 15 minutes). The sensor candidate is here SEVRI from METEOSAT. This method has very strong assets, 1 among many being the estimate of surface BRDF from previous day(s) and being re-injected into D-Day. Evaluation with satellite AOD inter-comparisons and validation with reference ground-based AERONET are achieved showing very convincing results.

This paper is overall excellent with very high quality results. Especially, I found it very well written, and therefore pleasant to read. I also very much appreciated the big efforts made by the authors to confront the results with the actual information content (variability) from SEVIRI, especially with respect (w.r.t.) geometry (via the scattering angle). This is little done nowadays while we all very well know this eventually plays a major role, especially with geostationary sensors.

I overall recommend this paper for publications. I only have the 4 following requests below (medium importance) and a few detailed questions in the next sub-section. I thank in advance Ceamanos *et al.* for addressing them:

1) Generally, we always expect different performance over water and land surfaces due to the different surface signal magnitude and how retrievals can be affected. Here, validations results are merged for all surfaces, but I believe it would be much more relevant to discriminate per surface type (AOD Land retrieval vs. AOD Ocean retrievals). I do anticipate that the number of AERONET station over waters is very low, and mostly coastal or islands. However, it is important to analyse in-depth whether iAERUS does better than MODIS over ocean (it is known nowadays MODIS AOD ocean is a little bit too high).

2) Could you please clarify how valid is the daily retrieval of surface BRDF over Ocean? I understand you assume the surface does not vary much over 1 day. I can largely see this assumption is true over Lands. Over oceans, I have some difficulties due to the potential high wind variabilities that might then lead to slope / wave changes (and hence glint). In addition, do you think we really need a daily Ocean BRDF from past day while I would assume we can easily calculate it instantaneously with the Cox & Munk model, Monahan + Koepke for foam / white caps, and wind forecast from meteo centres?

3) Could you give more details on the configuration for the BRDF surface retrieval over the past days? Notably, how do you select / filter out L1 pixels, what is the BRDF spatial resolution, and how aerosol effects are corrected for?

4) I miss references about the considered PODLER Aerosol versioning and its traceability. Could you please clarify it, given there have been several version I believe in the last versions.

**Details**

Abstract – Page 1: "of interest for research topics" => This is also for high importance for all operational and climate services, not only in research mode.

Sect 2.2.1 Page 6 "All aerosol terms depend on the aerosol optical depth ($\tau 56$, )" => They in fact depend on all aerosol properties (notably aerosol model and related scattering vs. absorption effects), not only AOD. Would you agree?

Sect 2.2.3 – Page 8: From this and Appendix A2.2., I slightly miss details on i) the foam fraction / white caps, and ii) the underlight scattering. My guess is for ii) you have followed the traditional Monahan & O'Muircheartaigh (1980) approach, and for ii) a typical standard Chrlorophy value (and perhaps even a fixed water leaving radiance). Could you please clarify these assumptions?

---

## Author Response (AR1)

**Answers to Anonymous Referee #1:**

*Summary:*
*Ceamanos et al. propose the new iAERUS-GEO retrieval algorithm, to derive aerosol optical depth (AOD) and associated "confidence measure" from SEVERI/MSG's full-disk observations every 15 minutes. Essentially, iAERUS-GEO assumes: fixed aerosol models, a Kalman filtered estimate of surface BRDF, and an optimal estimation method. Resulting AOD (from Jan 2012-Sept 2013 data) is evaluated by comparing to 1) ground-truth AERONET sunphotometers and 2) other retrievals from other sensors (e.g. GRASP/POLDER and NASA MODIS).*

*Overall assessment:*
*For this most part, the paper is logical and the technique and validations are explained well. I admit I got lost in the algorithm description (mathematics and weighting schematics) , but had a much easier time following the product description and comparisons with other datasets. The figures were high quality. I believe the data and analysis support the conclusions.*

We thank the reviewer for the valuable comments and the time spent reviewing our work. We have attentively addressed all the raised issues and have produced a revised version of the manuscript. Please find below our replies, the original comments by the reviewer are shown in italics. The line numbers given in our answers correspond to the new manuscript, with the "track changes" mode on.

*Specific comments:*
- *Lines 110-130: I think overall good use of bullets and flowchart*

Thanks.

- *In Figure 1: "apriori AOD from climatology": Is there a maximum allowable difference between final AOD and climatology?*

There is not any constraint on the difference between the two values of AOD. The inversion method allows the final AOD to depart more or less from the a priori AOD depending on the information content of the processed SEVIRI observations (quantified by the AOD Jacobian). This makes final AOD weakly (strongly) dependent on a priori AOD when information content is high (low).

- *Line 150-152: I believe it is sort of explained later, but the assumption of two layers only makes sense of both the Rayleigh optical depths and gas absorption optical depths are small. It would not work for a shorter wavelength (say, blue band) would it? Does the lower layer have any actual thickness (say, in km, or is it entirely arbitrary?). Connect with line 213, perhaps?*

The two-layer model was a simplified way to describe the atmosphere considered in iAERUS-GEO, which is composed of gases distributed throughout the atmosphere and an aerosol layer located near the surface, also containing gases (Fig. 3a). Step 1 of the algorithm corrects SEVIRI top-of-atmosphere observations for gas effects, thus Rayleigh scattering and gas absorption. This is done by the SMAC method, which considers a complete, multi-layer U.S. Standard

atmosphere to describe the vertical profiles of several gases including $O_2$, $CO_2$, $H_2O$, and $O_3$. At the end of Step 1, the atmosphere is defined by the aerosol layer only (Fig. 3b), which is exclusively described by its optical properties and therefore does not have any actual thickness (see new caption of Fig. 3). The correction for gas effects is done without taking into account the interaction between molecules and particles within the aerosol layer based on the fact that aerosol/molecular scattering coupling can be neglected for SEVIRI channel VIS06 (lines 230-232). This is the assumption that would not be valid at shorter wavelengths (see new sentence in line 672). To avoid further confusion, we have removed all the references to the 2-layer atmosphere in the manuscript. We also provide now further information on the U.S. standard atmosphere model used by SMAC in lines 224-227.

- *I like Fig 3, somehow the colors make it obvious what you mean. However, in the caption, does "gas effects" mean gas absorption or molecular scattering (or both?)*

"Gas (or molecular) effects" mean gas absorption and molecular scattering. This is made clearer in the new version of Fig. 3 (as well as the corresponding caption) and in lines 126 and 221.

- *Line 180: aerosol layer reflectance with error of 10% is actually quite large. Rule of thumb is 0.01 reflectance error = 0.1 AOD error. (depends on angle of course).*

Numerical calculations were done with the radiative transfer code ARTDECO to quantify the accuracy of TOL reflectance simulated by the MSA analytical model. Angles, aerosol, and surface parameters were set to common values found in SEVIRI observations. Results showed an increasing accuracy of MSA with scattering angle (*xi*), with an average relative error of ~10% for *xi*=60°, ~6% for *xi*=100°, ~3% for *xi*=140°, and ~2% for *xi*=180°. According to these results, the highest error of MSA of 10% happens only for low scattering angles and it is therefore counterbalanced by the fact that, in these geometries, satellite observations are fewer and aerosol content information is greater (see the resulting lower AOD errors in this case in Fig. 10b). Albeit the performances of MSA are perfectible, they were considered to be acceptable for processing SEVIRI data due to (1) the high speed of MSA and (2) the satisfactory results that are presented in the manuscript. Lines 189-194 were rewritten according to the above. Finally, we say in the conclusions that the radiative transfer modeling done in iAERUS-GEO (including MSA) will be improved for the future adaptation of iAERUS-GEO to FCI (lines 670-673)

- *Line 185: I don't really understand the logic of xi=30° being a cutoff point. What kind of "observing geometry" yields such a scattering angle?*

The cutoff angle was set to 30°, instead of 45° as it is done in Katsev et al., to avoid discarding satellite observations with scattering angles between these two values. Values lower than 45° are indeed uncommon but not impossible for GEO. For example, these geometries correspond to SEVIRI observations over the Caribbean (VZA=~70°), at sunset (SZA=~70°), and in the summertime (RAA=160-180°) (e.g., xi=40° if VZA=SZA=70° and RAA=180°). The discussion on the cutting angle has been moved to Appendix A.1.1, as it is a technical detail of the algorithm. There, we say now that observations with such a low scattering angle are seldom.

- *Line 197: I don't mind Maignan et al. model being chosen, but why?*

Maignan et al. (2004) proposes the modification of the Ross-Thick Li-Sparse (RTLS), or simply Ross-Li, model with the consideration of the back-scattering hot spot that can be often observed over some vegetation surfaces. This phenomenon can be seen from GEO orbit (Li et al., 2021). The addition of the hot spot therefore improves the RTLS model, which is widely used to characterize the land BRDF for the retrieval of Earth's (Lucht et al., 2000) and planetary (Ceamanos et al., 2011) surfaces, as well as for aerosol retrieval (Lyapustin et al., 2018). The Maignan's model is used in the GRASP aerosol algorithm for example (Dubovik et al., 2011). Part of this information was added in lines 210-213.

- *Page 9: I was completely lost here. I guess this mathematics is necessary, but I can't review it.*

Kalman filtering is a key element of iAERUS-GEO, as it helps providing a stable and accurate estimate of surface BRDF over the SEVIRI grid. Section 2.4.1 is therefore necessary while we admit that it is quite mathematical. We made an effort to make this section as simple as possible, and we refer the reader to the work of Carrer et al. (2010) for more details on the Kalman filter-based method. We now provide some extra information on the inversion over ocean (see lines 247-252).

- *Line 272: I feel that the benefits could be described?*

Benefits are now described in lines 295-296.

- *Line 274: Figure 4: What is the point of this figure?*

This figure shows the two sets of coefficients that are used to weight SEVIRI observations in the two consecutive inversions of the surface BRDF. Figure 4 is referred twice in Sect. 2.4.2.

- *Line 283-285: Discussion of super-pixels being consistent with Anderson et al. (2003) seems meaningless here. Especially if later on, we may be looking for plumes with much finer resolutions?*

References to Anderson et al. (2003) were removed from the manuscript. See answer to your comment #14 for a response on the subject of narrow plumes.

- *Line 297-313: Again, got lost in math here. Maybe a figure/diagram could help?*

Again, Sect. 2.5.2 is key to the algorithm and we did our best to explain the AOD inversion method as clear as possible. We refer the reader to the seminal work of Rodgers (2000) for more details on optimal estimation (OE), which is widely used in atmospheric sciences nowadays. We are currently working on another journal paper that will investigate the use of OE for aerosol retrieval from GEO, and will therefore provide further details on the method used in iAERUS-GEO.

- *Line 320/Figure 6: What if you could express confidence in that "I know darn sure that the AOD is low, I just can't tell you within %. However, I can tell you within ±0.03". This is why retrievals*

*such as MODIS tend to be expressed as ±(0.03 + 10%), including an absolute and relative component.*

Confidence measure (CM) is not meant to quantify the error of the retrieval but its robustness. This is possible thanks to its construction based upon the AOD Jacobian, which is directly linked to the information content of the satellite observations. Eventually, this makes CM to be correlated to the error of the retrieved AOD (see Fig. 10c). Nonetheless, it is important to note that low CM values do not necessarily correspond to low values of retrieved AOD as the reviewer seems to be suggesting. Indeed, low AOD Jacobians can happen for any AOD value as long as the surface brightness is close to the critical reflectance. Part of this information was included in line 345-346 of the revised manuscript.

- *Line 325-326 (spatial smoothing). What happens for example of a wildfire, which might have an extremely narrow plume? Would it get ignored?*

Spatial smoothing is indeed not appropriate in the presence of narrow, thick smoke (or dust) plumes. This is a limitation of the current version of iAERUS-GEO and should be improved in its future adaptation to FCI. One idea is to use the additional wavelengths of FCI to detect the presence of smoke or dust aerosols, so spatial smoothing can be disabled in this case (similar to what is done in MODIS/MAIAC for example). The detection of aerosol types was not attempted for SEVIRI due to the limited number of channels of this sensor. This is now explained in lines 360-361 and 669-670.

- *Line 327: Coastal: I don't understand this sentence.*

SEVIRI pixels along the coasts are not processed by iAERUS-GEO until the final step of spatial smoothing. The reason is the presence of land and water in the same pixel, which makes inappropriate the land and ocean BRDF models that are used in the algorithm. Instead, the average AOD value resulting from the 9x9-pixel centered box is ascribed to each coastal pixel. Lines 355-356 have been rewritten to explain this more clearly.

- *Line 338-341: I can't tell from this: Do the components matter? Or only the total AOD? If the components do matter, are they supposed to somehow connect with the Lyapustin-type maps discussed on next page?*

Total AOD from CAMS is used as a priori information, whereas component/species-related AOD from CAMS are used to calculate the monthly variation of the geographic distribution of Models 6 and 7. This is better explained now in Sect. 2.6.1 and line 410.

- *Line 377-379: How big a buffer? And are there buffers (mixing) between months – so as no sharp transitions ?*

Aerosol models are determined based upon the model maps such as the one in Fig.7. A 64x64-pixel box (aka. the buffer) is used to determine the predominant models for each pixel. Whereas only one model is available for most cases (e.g., Model 6 for the Sahara Desert in May, see Fig. 7), pixels near the transition among several models can have two or three models. The weight for each model is simply calculated by the number of pixels assigned to the model divided by the total number of pixels of the buffer. Weights are then used to linearly combine

the optical properties corresponding to each model. Currently, temporal buffers (from one month to another) are not considered but it is indeed a possible improvement for future versions of iAERUS-GEO. This is better explained now in lines 414-416.

- *Line 425-430: As I understand, GRASP retrieval ideas are somewhat similar in concept? In that you do a cube with an NxN box with T times? I feel that since you bring it up, maybe a user would be interested why wouldn't use a GRASP for your super-pixel retrievals?*

GRASP indeed incorporates a multi-pixel inversion technique, the main difference with iAERUS-GEO being the way the information from adjacent pixels is processed. In GRASP, the information of all pixels in the super-pixels is simultaneously considered in the inversion. In addition, additional inter-pixel smoothing terms are added to the inversion system. While being a very robust solution, this way of proceeding inevitably increases the complexity and the data dimensionality of the inversion, thus the computing time. Although Dubovik et al. (2011) claim that there are tools for optimizing the solution with such a structure, we decided to implement a simpler multi-pixel method in iAERUS-GEO that processes single pixels resulting from the weighted averaging of the so-called super-pixels. This results in a lower complexity of the inversion procedure and a higher processing speed, which is key for NRT processing. This information has been added in lines 317-319.

- *Line 438-9: This sentence makes no sense to me.*

This sentence has been rewritten (see lines 480-483)

- *Line 443- : Where does this 0.1° NEO data come from? And note that differences between Terra and Aqua are not simply 3 hours. (see Levy et al., [https://doi.org/10.5194/amt-11-4073-2018] for example).*

MODIS aerosol data were retrieved from the NASA Earth Observations website (https://neo.sci.gsfc.nasa.gov/). We successfully used these data in the past (e.g. Ceamanos et al., 2021), which have the advantage of being available at a higher spatial resolution (0.1°) than other MODIS aerosol products (usually at 1°). The differences between Aqua and Terra aerosol products are well noted but are not relevant in our study as only Terra is used in Section 4.5.

- *Line 455-457: I can understand why there might be a larger "absolute" bias, but why do the lower sensitivities over bright surfaces lead to a high positive bias?*

The positiveness of MBE along the dust belt is likely related to systematic biases of the MSA method, which was found to underestimate the satellite reflectance in the case of bright surfaces and Model 6 (desert aerosols). This gives place to the systematic AOD overestimation over desert sites that can be observed in Fig. 9c. This is discussed in the text now (lines 502-504).

- *Figure 9: panel (a): What is the weird stripe of missing data at y = 1.6? (c) because I can't see white edge/black edge, suggest doing >MBE and <MBE with, say blue      red color scale for example (0.0 as white in middle – you do it in Fig 12)*

After double-checking, the apparent scarcity of AERONET AOD data at y=1.6 is purely fortuitous. The plot showing MBE has been regenerated as suggested by the reviewer.

- *Figure 10: I cannot read the grey text boxes in each panel – there is room to make bigger.*

Legend boxes have been made bigger in all figures of the article.

- *Figure 11: See my comment #13. I don't necessarily like you screening out the cases of CM=1 and CM=2, because that neglects low AOD cases.*

As said previously (see answer to your comment #13), low CM values do not necessarily correspond to low AOD estimates. On the contrary, the average AOD retrieved by iAERUS-GEO decreases with an increasing CM filtering because the lowest information content usually happens over bright surfaces where aerosol load is higher. In particular, the average retrieved AOD is [0.17, 0.15, 0.15, 0.14, 0.09] for cases [CM>1, CM>2, CM>3, CM>4, CM>5] respectively, as it is now shown in the new version of Fig. 11 and in lines 532-534.

- *Fig 13: (a) My eyes can't help see the "random values" of iAERUS-GEO AOD, especially when AERONET AOD is down near zero. And that (b) GRASP looks so much less random! Of course, it's information content.*

Indeed, iAERUS-GEO shows an apparently higher random noise (i.e., points departing from the 1:1 line) despite having a slightly lower error in average (RMSE is 0.093 for iAERUS-GEO and 0.102 for GRASP; Fig. 13). We believe that one main reason to this is that there are 55 times more points for iAERUS-GEO than for GRASP, which obviously increases the chances of "seeing" random noise. RTM biases can also contribute to this noise. Finally, we would like to note that the spurious line at y~=0.05 in Fig. 13a comes from cloud contaminated pixels that were not correctly detected in the input cloud mask (see new comment in line 499).

- *Line 525 (and Figure 14). That is so interesting about the scattering angles across the disk.*

Thanks.

- *Fig 15: Pretty compelling time series matchups overall. I do note however, that, say in (a = Tenerife) that for every case where iAERUS retrieves high values (cloud contamination), there are also uncanny low retrievals (when AERONET says it should be high – 03 Aug, 11 Aug).*

AOD overestimation on 7 and 17 August comes indeed from cloud contaminated pixels (see lines 592-593). AOD underestimation on 3 and 11 August is likely to come from a lower accuracy of the RTM at high zenith angle and high AOD (note that uncanny low retrievals happen in the morning). This limitation is mentioned now in line 671.

- *Skip to conclusions: I will definitely be curious how the 2-layer assumption will be valid if going to shorter wavelengths on FCI/MTG.*

Please refer to our answer to your comment #3 for a discussion on the 2-layer atmosphere. The assumption of no molecular/particular coupling will not be valid for the shorter wavelengths

from FCI. This will be circumvented in the adaptation of iAERUS-GEO to FCI by integrating "step 1" (dealing with gas correction) into "step 3" to estimate AOD directly from TOA reflectances while accounting for gas effects (see Fig. 1 for details on the iAERUS-GEO main steps). This is now discussed in line 672.

- *Appendices: I did not check these in detail.*

Thanks again for the time devoted to reviewing our manuscript.

Xavier Ceamanos

**Answers to Anonymous Referee #2:**

*General*

*Ceamanos et al. have developed and extended the iAERUS-GEO retrieval algorithm to retrieve Aerosol Optical Depth (AOD) at a very high temporal frequency (every 15 minutes). The sensor candidate is here SEVRI from METEOSAT. This method has very strong assets, 1 among many being the estimate of surface BRDF from previous day(s) and being re-injected into D-Day. Evaluation with satellite AOD inter-comparisons and validation with reference ground-based AERONET are achieved showing very convincing results.*

*This paper is overall excellent with very high quality results. Especially, I found it very well written, and therefore pleasant to read. I also very much appreciated the big efforts made by the authors to confront the results with the actual information content (variability) from SEVIRI, especially with respect (w.r.t.) geometry (via the scattering angle). This is little done nowadays while we all very well know this eventually plays a major role, especially with geostationary sensors.*

*I overall recommend this paper for publications. I only have the 4 following requests below (medium importance) and a few detailed questions in the next sub-section. I thank in advance Ceamanos et al. for addressing them:*

We thank the reviewer for the valuable comments and the time spent reviewing our work. We have attentively addressed all the raised issues and have produced a revised version of the manuscript. Please find below our replies, the original comments by the reviewer are shown in italics. The line numbers given in our answers correspond to the new manuscript, with the "track changes" mode on.

1) *Generally, we always expect different performance over water and land surfaces due to the different surface signal magnitude and how retrievals can be affected. Here, validations results are merged for all surfaces, but I believe it would be much more relevant to discriminate per surface type (AOD Land retrieval vs. AOD Ocean retrievals). I do anticipate that the number of AERONET station over waters is very low, and mostly coastal or islands. However, it is important to analyse in-depth whether iAERUS does better than MODIS over ocean (it is known nowadays MODIS AOD ocean is a little bit too high).*

In this work, validation was performed using the SEVIRI pixels containing each AERONET site (line 458). This makes all iAERUS-GEO retrievals to correspond to land (said now in lines 27 and 649). Assessment of retrieved AOD over ocean using AERONET data can be done by (1) using large pixel boxes to get a sufficient number of pure ocean pixels and (2) discarding the pixels which are adjacent to land (e.g. Chen et al., who used 90×90 km boxes). Unfortunately, we could not adopt this methodology in our study as the 18-month AOD retrievals were made available over limited regions surrounding each ground station to avoid the huge computational resources that would be required to process the full-disk data for such a long period of time. A separated land/water full assessment is planned to be done in the near future when the first collection of iAERUS-GEO/SEVIRI full-disk data will be available (see new comment in line 654).

2) *Could you please clarify how valid is the daily retrieval of surface BRDF over Ocean? I understand you assume the surface does not vary much over 1 day. I can largely see this assumption is true over Lands. Over oceans, I have some difficulties due to the potential high wind variabilities that might then lead to slope / wave changes (and hence glint). In addition, do you think we really need a daily Ocean BRDF from past day while I would assume we can easily calculate it instantaneously with the Cox & Munk model, Monahan + Koepke for foam / white caps, and wind forecast from meteo centres?*

Indeed, surface BRDF varies more rapidly over ocean than over land. The variations of water reflectance due to the changing winds are taken into account in the instantaneous retrieval of AOD by using 15-min interpolated 3-h winds from ECMWF ERA5 (lines 438-440). These input data allow us to calculate the fractional cover of whitecaps ($f_{wc}$) following Monahan and Muircheartaigh (1980) and the glint reflectance ($f_1^o$) following Cox and Munk (1954). The Kalman filtering in the daily BRDF retrieval is mainly aimed at retrieving the Lambertian component ($k_0^o$), which for the most part depends on water leaving radiance and which is therefore assumed to vary more slowly with time. The reviewer helped us spotting a mistake in the manuscript, as the anisotropic sun-glint BRDF coefficient $k_1^o$ is indeed not estimated by the Kalman filter and it is simply obtained by doing $1 - f_{wc}$. All this was made clearer in Sect. 2.4.1 (lines 247-252) and Appendix A.2.2 (see lines 762-764).

3) *Could you give more details on the configuration for the BRDF surface retrieval over the past days? Notably, how do you select / filter out L1 pixels, what is the BRDF spatial resolution, and how aerosol effects are corrected for?*

SEVIRI observations are filtered in iAERUS-GEO at the very beginning of the processing in order to discard data with unwanted geometry (i.e. in the sun glint region, with scattering angle lower than the cutting angle, and corresponding to high zenith angles) and contaminated by clouds or snow (see now lines 442-450). The resulting SEVIRI observations are corrected for gas effects in step 1 to be then all used in step 2 for daily BRDF estimation and step 3 for 15-min AOD estimation (see Fig. 1). Surface BRDF is estimated at the native SEVIRI grid (see line 238), with a highest spatial resolution of 3 km at the sub-satellite point. Surface BRDF is estimated by compensating for aerosol effects through the estimation of the daily-average AOD ($\tau_{daily}$), as it was done in the original AERUS-GEO algorithm (see line 246 for the state parameters to be inverted and containing the daily AOD).

4) *I miss references about the considered PODLER Aerosol versioning and its traceability. Could you please clarify it, given there have been several version I believe in the last versions.*

GRASP/POLDER data correspond to the version "Models", version 2.1, and Level-2 (see lines 473-474). The website from which the data were downloaded has been updated in the manuscript for the sake of completeness to https://www.grasp-open.com/products/polder-data-release/.

**Details**
*Abstract – Page 1: "of interest for research topics" => This is also for high importance for all operational and climate services, not only in research mode.*

This has been corrected in line 18.

*Sect 2.2.1 Page 6 "All aerosol terms depend on the aerosol optical depth ($\tau$56, )" => They in fact depend on all aerosol properties (notably aerosol model and related scattering vs. absorption effects), not only AOD. Would you agree?*

Yes, we agree. However, the point here is to make clear that all these terms depend on the variable that we want to estimate (i.e. AOD). This has been made clearer in line 167.

*Sect 2.2.3 – Page 8: From this and Appendix A2.2., I slightly miss details on i) the foam fraction / white caps, and ii) the underlight scattering. My guess is for ii) you have followed the traditional Monahan & O'Muircheartaigh (1980) approach, and for ii) a typical standard Chrlorophy value (and perhaps even a fixed water leaving radiance). Could you please clarify these assumptions?*

Indeed, the fractional cover of whitecaps ($f_{wc}$) is calculated following Monahan and Muircheartaigh (1980). Underlight reflectance is calculated from the satellite data through the estimation of the Lambertian component ($k_0^o$) with the Kalman filter approach. See my answer to your comment #2 for more details.

Thanks again for the time devoted to reviewing our manuscript.

Xavier Ceamanos